# Trimethylamine N-oxide impairs β-cell function and glucose tolerance

Lijuan Kong [1,2,3,7], Qijin Zhao [1,2,3,7], Xiaojing Jiang[1,2,3,7], Jinping Hu[1], Qian Jiang[1,2,3], Li Sheng[1], Xiaohong Peng[4,5], Shusen Wang [6], Yibing Chen[1,2,3], Yanjun Wan[1,2,3], Shaocong Hou [1,2,3], Xingfeng Liu [1,2,3], Chunxiao Ma[1,2,3], Yan Li[1], Li Quan[5], Liangyi Chen [4,5], Bing Cui [1,3] & Pingping Li [1,2,3] ✉

β-Cell dysfunction and β-cell loss are hallmarks of type 2 diabetes (T2D). Here, we found that trimethylamine N-oxide (TMAO) at a similar concentration to that found in diabetes could directly decrease glucose-stimulated insulin secretion (GSIS) in MIN6 cells and primary islets from mice or humans. Elevation of TMAO levels impairs GSIS, β-cell proportion, and glucose tolerance in male C57BL/6 J mice. TMAO inhibits calcium transients through NLRP3 inflammasome-related cytokines and induced Serca2 loss, and a Serca2 agonist reversed the effect of TMAO on β-cell function in vitro and in vivo. Additionally, long-term TMAO exposure promotes β-cell ER stress, dedifferentiation, and apoptosis and inhibits β-cell transcriptional identity. Inhibition of TMAO production improves β-cell GSIS, β-cell proportion, and glucose tolerance in both male *db/db* and choline diet-fed mice. These observations identify a role for TMAO in β-cell dysfunction and maintenance, and inhibition of TMAO could be an approach for the treatment of T2D.

Type 2 diabetes (T2D) is a global health problem, and its prevalence is increasing steadily each year. T2D is a chronic and progressive disease characterized by insulin resistance and/or insufficient insulin secretion resulting from β-cell dysfunction and decreases in β-cell mass[1–3]. Although both factors play a role, decreased β-cell function and β-cell mass are the predominant factors for progression to T2D[3]. Pioneering studies have shown that β-cell function is already reduced to 50 to 80% of normal at the time of T2D onset[4]. Reduced β-cell function, manifested as decreased glucose-stimulated insulin secretion (GSIS), is a prerequisite for the progression from normal glucose tolerance to hyperglycemia[5–7]. Pancreatic β-cells secrete insulin in a biphasic manner, first- and second-phase insulin secretion, and first-phase insulin release is almost lost in patients with impaired glucose tolerance or in the early stages of T2D[8]. Mounting evidence suggests that the loss of β-

cell function in stressed islets results not only from β-cell death but also from changes in the expression of genes that define β-cell identity and state of differentiation[9–11].

The gut microbiota is associated with a variety of diseases, including obesity, insulin resistance, diabetes, and nonalcoholic fatty liver disease (NAFLD). Changes in the human gut microbiota impact β-cell function[12]. Dietary factors, such as choline, phosphatidylcholine and L-carnitine, can be metabolized by gut microbes to produce trimethylamine (TMA). TMA is then carried via the portal circulation to the liver, where it is rapidly converted by host hepatic flavin monooxygenase 3 (FMO3) into trimethylamine N-oxide (TMAO)[13]. Diabetes is associated with higher TMAO plasma levels in mice and humans[14,15]. In insulin-resistant mice fed a high-fat diet (HFD) or in leptin knockout mice, elevation of TMAO through diet significantly exacerbated insulin

[1]State Key Laboratory of Bioactive Substance and Function of Natural Medicines, Institute of Materia Medica, Chinese Academy of Medical Sciences and Peking Union Medical College, Beijing, China. [2]Diabetes Research Center of Chinese Academy of Medical Sciences, Beijing, China. [3]CAMS Key Laboratory of Molecular Mechanism and Target Discovery of Metabolic Disorder and Tumorigenesis, Beijing, China. [4]College of Future Technology, Institute of Molecular Medicine, National Biomedical Imaging Center, Peking University, 100871 Beijing, China. [5]Beijing Key Laboratory of Cardiometabolic Molecular Medicine, Peking University, 100871 Beijing, China. [6]Tianjin First Central Hospital, Tianjin, China. [7]These authors contributed equally: Lijuan Kong, Qijin Zhao, Xiaojing Jiang. ✉e-mail: lipp@imm.ac.cn

resistance in these mice[16,17]. Knockdown or inhibition of Fmo3, the TMAO-producing enzyme, prevented insulin resistance in liver insulin receptor knockout (LIRKO) mice and inhibited obesity in HFD-fed mice[18,19]. Whether TMAO at a pathological dose directly impairs β-cell function in vivo remains unclear.

Here, we reported that TMAO inhibited β-cell GSIS, promoted β-cell dedifferentiation and apoptosis and repressed transcriptional identity. Short-term TMAO stimulation decreased oxidative phosphorylation (OXPHOS) and ATP production, reduced sarco/endoplasmic reticulum ATPase type 2 (Serca2) through NLR family pyrin domain containing 3 (NLRP3) inflammasome-related inflammatory cytokines, and subsequently inhibited glucose-stimulated calcium transients. Long-term TMAO treatment resulted in ER stress and β-cell

dedifferentiation. A reduction in the TMAO concentration by knockdown of Fmo3 improved GSIS and the β-cell proportion in both diabetic *db/db* mice and choline diet-fed mice. Together, these findings suggest that the gut microbiota metabolite TMAO/FMO3 might be a valuable drug target for the treatment of β-cell dysfunction and T2D.

## Results

### TMAO had elevated levels in diabetic mice and subjects with diabetes and inhibited GSIS in islets

We measured TMAO levels and found that the TMAO level was increased 2.5-fold in 9-week-old leptin receptor-deficient diabetic (*db/db*) mice (Fig. 1a). Both the RNA and protein levels of Fmo3 were also largely increased in the livers of the *db/db* mice compared with the

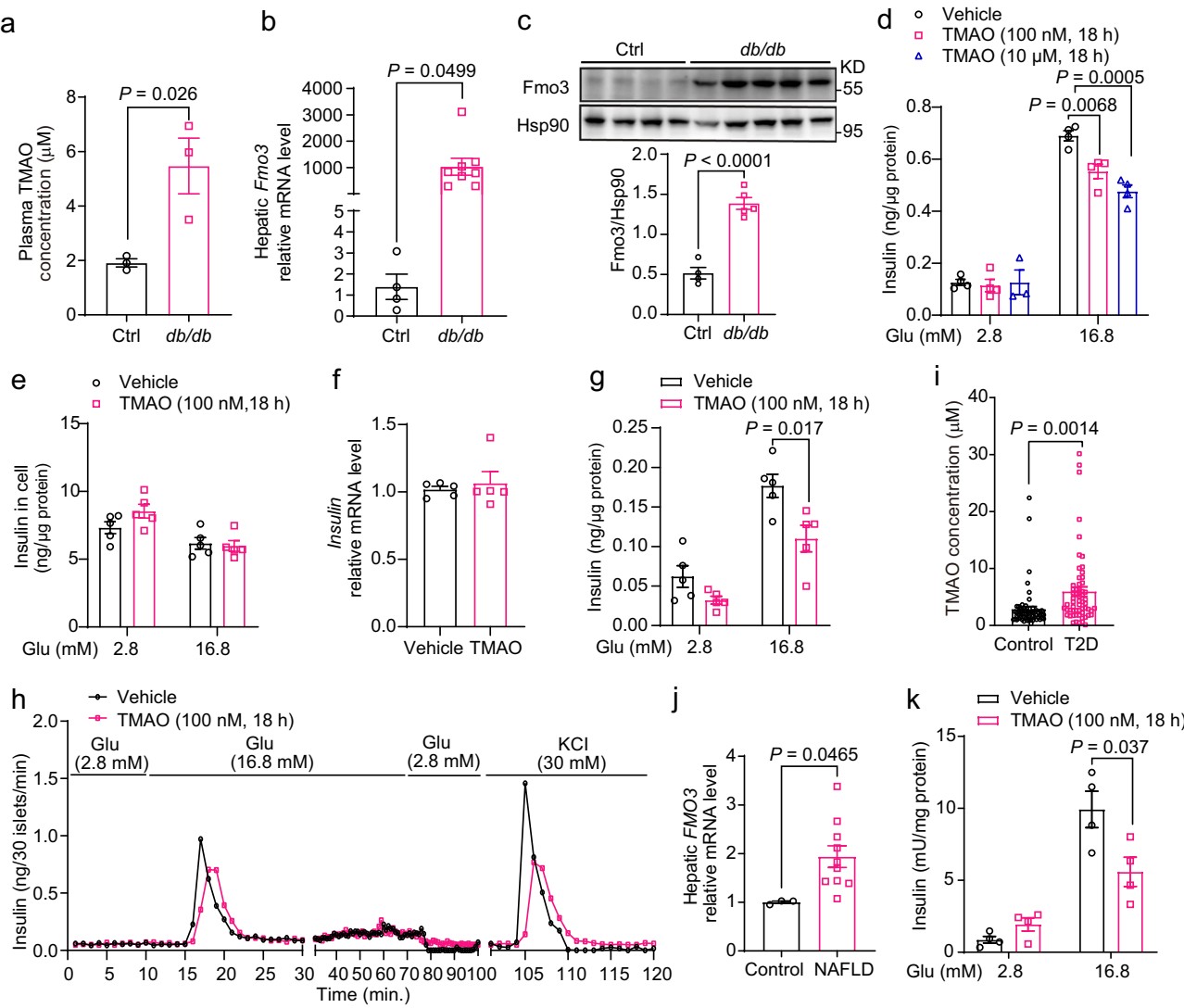

**Fig. 1 | TMAO was elevated in diabetic mice and subjects with diabetes and inhibited glucose-stimulated insulin secretion in islets. a** Plasma TMAO concentration in 9-week-old male control and *db/db* mice. *n* = 3 mice. **b** Hepatic *Fmo3* mRNA levels in 15-week-old male control (*n* = 4 mice) and *db/db* mice (*n* = 8 mice). **c** Hepatic Fmo3 protein levels in 15-week-old male control (*n* = 4 mice) and *db/db* mice (*n* = 5 mice). **d** GSIS in MIN6 cells treated with or without TMAO (100 nM and 10 μM, 18 h). *n* = 3 (10 μM TMAO at 2.8 mM Glu) or 4 (others) biologically independent cell samples. **e** Intracellular insulin levels in MIN6 cells treated with or without TMAO (100 nM) for 18 h during GSIS. *n* = 5 biologically independent cell samples. **f** Intracellular *insulin* mRNA levels in MIN6 cells treated with or without TMAO (1 μM) for 24 h. *n* = 5 biologically independent cell samples. **g** GSIS in mouse primary islets treated with or without TMAO (100 nM, 18 h). *n* = 5 biologically

independent islet samples. **h** Insulin levels measured during the perifusion of isolated mouse islets. *n* = 3 independent experiments. The levels of insulin secretion were measured after adding 2.8 mM glucose and 16.8 mM glucose during perifusion. The addition of 30 mM KCl led to membrane depolarization, which increased insulin secretion. **i** Serum TMAO concentration in control and T2D subjects. *n* = 60 subjects. **j** Hepatic *FMO3* mRNA levels in control (*n* = 3 subjects) and NAFLD subjects (*n* = 10 subjects). **k** GSIS in human primary islets treated with or without TMAO (100 nM, 18 h). *n* = 4 biologically independent samples. Statistical significance was calculated (**a**–**d**, **g**, **i**–**k**) by two-sided Student's *t*-test. The data are presented as mean ± SEM. Mice: C57BL/6 J, 12 weeks old (**g**, **h**). Glu, glucose. Source data are provided as a Source Data file.

control mice (Fig. 1b, c), with similar *Fmo3* mRNA levels in other tissues (Supplementary Fig. 1a). The *db/db* mice had severely impaired glucose tolerance and reduced GSIS and C-peptide levels at 10 min after an oral glucose challenge (Supplementary Fig. 1b–d). Interestingly, in HFD-fed mice, which are insulin resistant but not diabetic, plasma TMAO levels (22 weeks of HFD feeding) were decreased rather than increased (Supplementary Fig. 1e). Similarly, hepatic *Fmo3* mRNA and pancreatic TMAO levels were unchanged or decreased (19 weeks of HFD feeding, Supplementary Fig. 1f, g). This finding suggested that TMAO and Fmo3 levels are elevated in diabetes.

Next, we explored the direct effect of TMAO on β-cell GSIS. MIN6 cells, mouse-derived insulin-secreting β-cells, were treated with or without TMAO (100 nM or 10 μM), followed by GSIS experiments. The results showed that 12 h and 18 h of TMAO treatment substantially inhibited GSIS (-30%, Fig. 1d and Supplementary Fig. 1h), with no change in cell viability (Supplementary Fig. 1i), intracellular insulin content (Fig. 1e), or *insulin* mRNA levels (Fig. 1f). Additionally, TMAO slightly inhibited arginine- but not GLP-1-stimulated insulin secretion (Supplementary Fig. 1j). In isolated primary islets from C57BL/6 J mice, TMAO treatment (100 nM, 18 h) significantly inhibited insulin secretion by glucose and KCl (Fig. 1g, h). Consistently, the GSIS response in the islet perifusion experiment curves revealed that insulin secretion in the first phase was strongly inhibited by TMAO (Fig. 1h). Moreover, TMAO (100 nM, 18 h) had no effect on ER stress, as reflected by p-PERK, p-eIF2α, p-IRE1, XBP1s and ATF6 expression in MIN6 cells (Supplementary Fig. 1k).

Importantly, the serum TMAO concentration in the T2D subjects was 2-fold higher than that in the controls (Fig. 1i). Hepatic *FMO3* mRNA levels were doubled in the NAFLD subjects compared to the controls (Fig. 1j). Consistently, GSIS in human primary islets was reduced by approximately 40% with TMAO (100 nM, 18 h) treatment (Fig. 1k).

Together, these results indicated that TMAO levels were elevated in diabetes and impaired β-cell function in islets from both mice and human subjects.

### Elevation of TMAO levels by dietary choline feeding impaired GSIS and β-cell maintenance in C57BL/6 J mice

To determine whether these in vitro results translated to in vivo conditions, we fed C57BL/6 J mice a chow diet (0.08% choline) or a choline diet (1% choline, Fig. 2a), which increases the plasma TMAO content[20]. The body weight and epididymal white adipose tissue (epi-WAT) weight of the choline diet-fed mice were slightly higher (Supplementary Fig. 2a, b), with no significant difference in food intake between the two groups of mice (Supplementary Fig. 2c). After 4 weeks of choline diet feeding, the plasma TMAO concentration in the choline group reached 18.9 μM, which was higher than that in the controls (2.3 μM, Fig. 2b). The choline diet-fed mice became glucose intolerant (Fig. 2c), with impaired GSIS reflected by the almost complete blockade of insulin secretion at 2 min after the glucose challenge (Fig. 2d, e). Similarly, the increase in plasma C-peptide levels 2 min after glucose stimulation was also reduced in the choline diet-fed mice (Fig. 2f, g). However, the intraperitoneal injection insulin tolerance test (IPITT) showed that insulin sensitivity in the two groups was similar (Supplementary Fig. 2d). Furthermore, we performed a hyperglycemic clamp, the gold standard for evaluating pancreatic islet function, in these mice. Blood glucose and the glucose infusion rate (GIR) in the two groups of mice were comparable (Supplementary Fig. 2e, f), indicating no change in insulin sensitivity. However, the plasma insulin level of the choline diet-fed mice during the hyperglycemic clamp was significantly lower than that of the control mice, with a 60% reduction in the area under the curve (AUC) of first-phase insulin secretion and a 24% reduction in the AUC of second-phase insulin secretion (Fig. 2h–j). Consistently, GSIS was also impaired in islets isolated from the choline diet-fed mice compared to those from the controls (Fig. 2k). These

results indicated that the choline diet impaired β-cell function and glucose tolerance but not insulin sensitivity in C57BL/6 J mice.

Next, we evaluated the effect of a choline diet on islet morphology in these mice. Hematoxylin-eosin (HE) staining showed that the islets of the choline diet-fed mice exhibited irregular islet shapes, and the islets were atrophied; the islet cells had various sizes and shapes, the boundaries were unclear, the arrangement was disordered, and the number was evidently reduced; some of the nuclei were pyknotic and divided (Fig. 2l). Moreover, we observed inflammatory cell infiltration in islets from the mice fed a choline diet (Fig. 2l), and the expression of inflammatory genes, such as *Tnf* and *Adgre1*, in the pancreas from the choline group was notably increased (Supplementary Fig. 2g). Strikingly, immunofluorescence showed that insulin staining decreased by 45% and glucagon staining increased by 3 times (Fig. 2m–o), along with higher plasma glucagon and α-cell mass and lower β-cell mass (Fig. 2p and Supplementary Fig. 2h, i). Contrary to the findings in MIN6 cells, TMAO had no effect on glucagon secretion in the α cell line αTC1-6 (Supplementary Fig. 2j).

### Reduction in TMAO levels by genetic deletion of Fmo3 in C57BL/6 J mice augmented insulin secretion

To further confirm that TMAO impairs β-cell function and glucose tolerance, we examined global Fmo3 knockout (*Fmo3⁻/⁻*) mice generated through CRISPR–Cas9-mediated gene editing[19] and fed them a choline diet (Fig. 3a). Hepatic *Fmo3* mRNA levels were decreased by ~80% in the *Fmo3⁻/⁻* mice, with an ~60% reduction in Fmo3 protein levels (Fig. 3b and Supplementary Fig. 3a). Consistently, plasma TMAO levels were reduced by ~80% (Fig. 3c), whereas there were no differences in food intake, body weight or tissue weight between the groups (Supplementary Fig. 3b–d). The *Fmo3⁻/⁻* mice exhibited improved glucose tolerance during the intraperitoneal injection glucose tolerance test (IPGTT) after 9 weeks of choline diet feeding (Fig. 3d). These two groups exhibited similar insulin sensitivity, as shown by insulin tolerance tests (ITTs) (Supplementary Fig. 3e). In addition, the increases in plasma insulin and C-peptide levels 2 min after intraperitoneal injection of glucose were improved in the *Fmo3⁻/⁻* choline diet-fed mice (Fig. 3e, f). Moreover, the hyperglycemic clamp further proved that the plasma insulin levels of the *Fmo3⁻/⁻* choline diet-fed mice 1 min after glucose injection through the jugular vein were largely increased compared to those of the control mice (Fig. 3g, h). Blood glucose and GIR in the two groups of mice were comparable (Supplementary Fig. 3f, g), indicating no change in insulin sensitivity. Consistently, Fmo3 knockout led to 40% increased insulin staining in the pancreas, along with elevated β-cell mass, indicating improved β-cell reduction (Fig. 3i–k and Supplementary Fig. 3h, i). Collectively, these data showed that Fmo3 deficiency in choline-fed mice improved islet morphology, β-cell function and glucose intolerance.

### TMAO inhibited glucose-stimulated ATP production and cytosolic calcium transients

Generally, glucose enters β-cells and yields ATP through glycolysis and the tricarboxylic acid cycle (TCA). The relative increase in the ratio of ATP to ADP leads to the closure of the $K_{ATP}$ channel. In turn, this change activates voltage-dependent calcium channels (VDCCs) to allow the influx of calcium that constitutes the triggering pathway required for insulin secretion. TMAO did not alter the glucose transporter *Glut2* mRNA level or glucose uptake in MIN6 cells (Supplementary Fig. 4a, b), whereas it significantly decreased the ATP content in MIN6 cells and primary islets from C57BL/6 J mice (Fig. 4a, b, and Supplementary Fig. 4c). Further studies showed that TMAO attenuated mitochondrial OXPHOS and promoted glycolysis, as shown by the oxygen consumption rate (OCR, Fig. 4c) and extracellular acidification rate (ECAR, Fig. 4d) curves, which contributed to the obvious decrease in ATP levels in MIN6 cells. The calculated OCR and ECAR curves

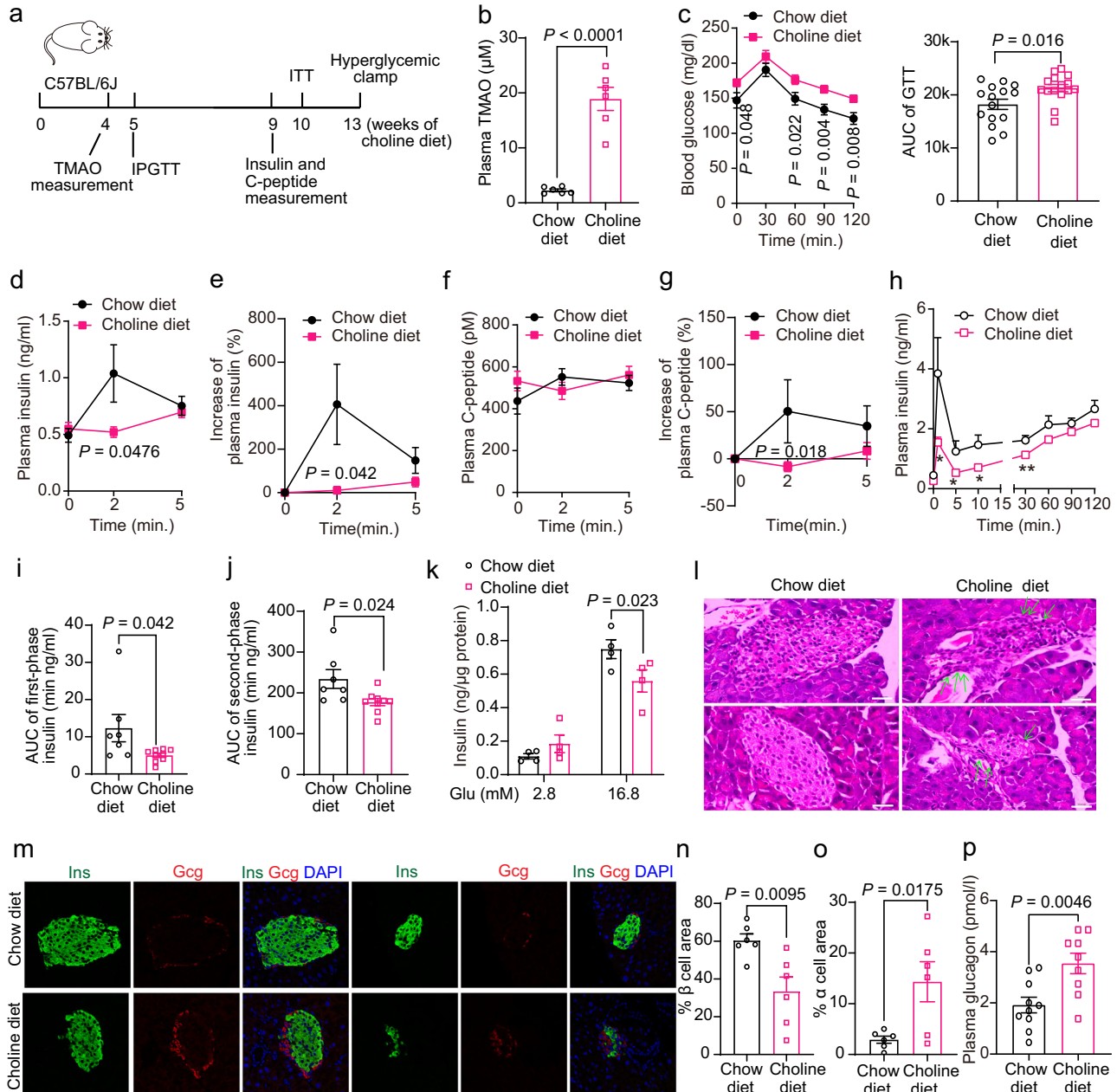

**Fig. 2 | A choline diet elevated plasma TMAO levels and impaired β-cell function and β-cell maintenance in C57BL/6 J mice. a** Schematic diagram of tests in male mice during chow diet and choline diet feeding. **b** Plasma TMAO concentration of male chow- and choline diet-fed (4 weeks) mice. $n = 6$ mice. **c** IPGTT of chow- and choline diet-fed (5 weeks) mice. $n = 15$ mice. The glucose dose was 2 g/kg of body weight. This step was followed by determining the AUC of the GTT. **d–g** Plasma insulin (**d**), increase in plasma insulin (%, **e**), C-peptide (**f**) and increase in C-peptide (%, **g**) in male chow- and choline diet-fed (9 weeks) mice after 0, 2, and 5 min of intraperitoneal injection of glucose. $n = 15$ mice (**d**, **e**); $n = 7$ mice (**f**, **g**). **h–j** Plasma insulin and AUC of first-phase (0–5 min) and second-phase (5–120 min) insulin levels during hyperglycemic clamp of male chow- ($n = 7$ mice) and choline diet-fed (13 weeks) mice ($n = 9$ mice). **k** GSIS of primary islets from male chow- and choline

diet-fed (10 weeks) mice. $n = 4$ biologically independent islet samples. **l** Pancreatic HE staining in male chow- and choline diet-fed (13 weeks) mice. The arrowhead indicates inflammatory cells. Scale bar, 100 μm. **m–o** Immunofluorescence of insulin (green), glucagon (red), and DAPI (blue) in paraffin-embedded pancreas sections from male chow- and choline diet-fed mice (**m**). This analysis was followed by measurements of % β-cell area (**n**) and % α cell area (**o**). $n = 6$ mice. Scale bar, 20 μm. **p** Plasma glucagon levels of male chow- ($n = 10$ mice) and choline diet-fed (6 weeks) mice ($n = 9$ mice). Statistical significance was calculated (**b–e**, **g–k**, **n–p**) by two-sided Student's $t$-test. $P$ values in (**h**) denoted by asterisks (from left to right): $P = 0.049$, $P = 0.046$, $P = 0.026$, $P = 0.006$. Mice: C57BL/6 J, choline diet feeding from 8 weeks old (**a–p**). Ins, insulin; Gcg, glucagon; AUC, area under the curve. Source data are provided as a Source Data file.

suggested that TMAO remarkably inhibited mitochondrial spare respiratory capacity (Fig. 4e) and increased glycolytic capacity (Fig. 4f). These results indicated that mitochondrial function was impaired after TMAO treatment.

Compared with the control, TMAO (100 nM, 18 h) treatment obviously increased the average resting $[Ca^{2+}]_c$ detected by Fura-2 (Fig. 4g) in low-glucose (2.8 mM) KRHB medium in MIN6 cells. In

primary islets from GCaMP6f mice, which expressed the genetically encoded calcium indicator GCaMP6f in β cells and were used for the dynamic measurement of cytosolic calcium in β cells, TMAO (100 nM, 18 h) treatment significantly reduced the peak value of calcium transients in normal β cells (Fig. 4h, i). Similarly, in the TMAO group, the peak value of cytosolic calcium transients induced by high glucose (16.8 mM) was inhibited in MIN6 cells overexpressed with GCaMP6f

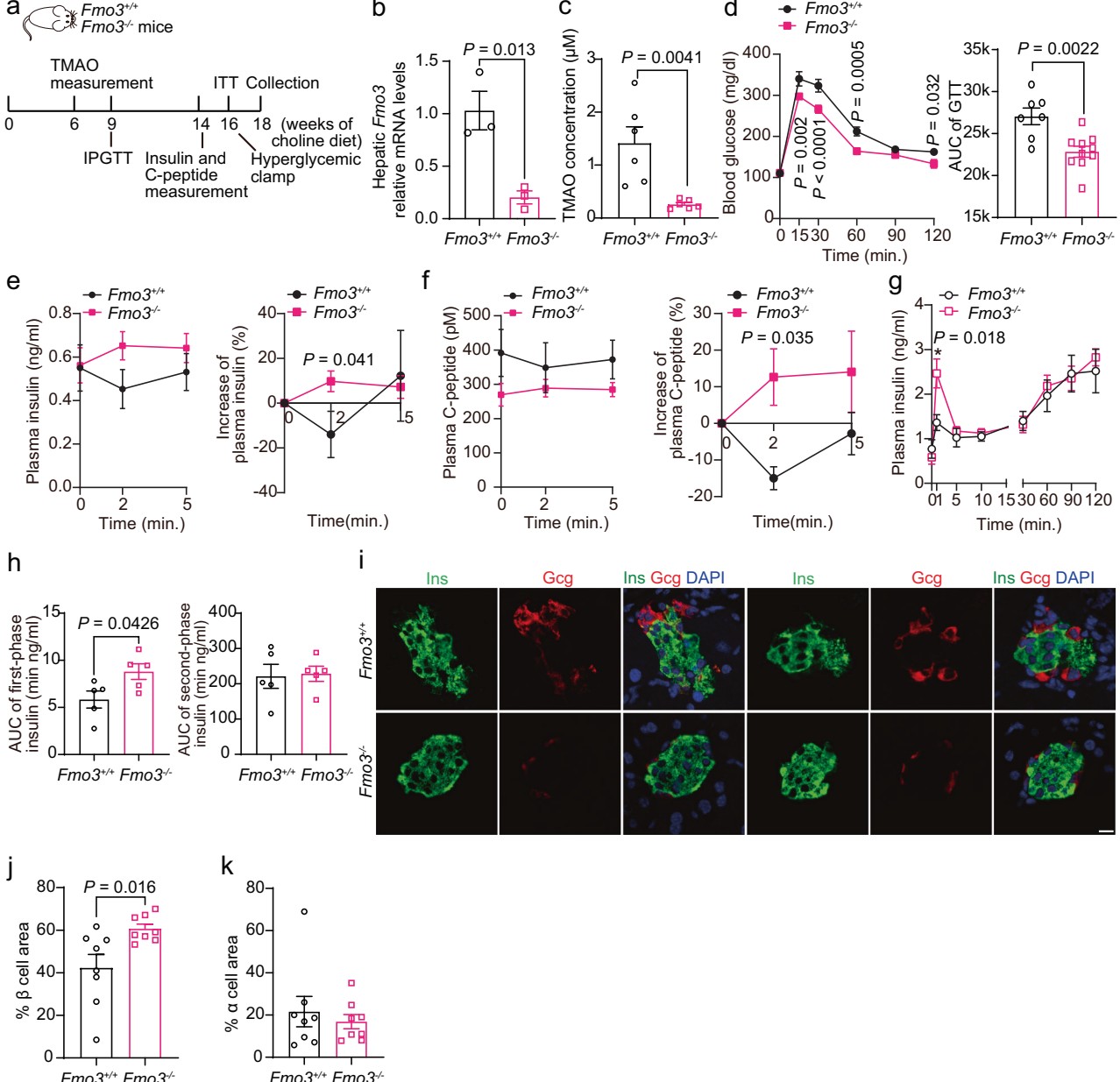

**Fig. 3 | Genetic knockdown of Fmo3 improved β-cell function, β-cell loss and glucose homeostasis in choline diet-fed mice. a** Schematic diagram of tests in male *Fmo3*[+/+] and *Fmo3*[-/-] mice fed a choline diet. **b** Hepatic *Fmo3* mRNA levels in male *Fmo3*[+/+] and *Fmo3*[-/-] mice. *n* = 3 mice. **c** Plasma TMAO levels in male *Fmo3*[+/+] and *Fmo3*[-/-] mice after 6 weeks of a choline diet. *n* = 6 mice. **d** IPGTTs of male choline diet-fed (9 weeks) *Fmo3*[+/+] (*n* = 7 mice) and *Fmo3*[-/-] mice (*n* = 10 mice). Then, the AUC of the GTT was determined. **e** Plasma insulin levels and increases in plasma insulin (%) after intraperitoneal injection of glucose for 0, 2, and 5 min in male choline diet-fed (14 weeks) *Fmo3*[+/+] and *Fmo3*[-/-] mice. *n* = 7 mice. **f** Plasma C-peptide levels and increases in plasma C-peptide (%) after intraperitoneal injection of

glucose for 0, 2, and 5 min in male choline diet-fed (15 weeks) *Fmo3*[+/+] (*n* = 6 mice) and *Fmo3*[-/-] mice (*n* = 10 mice). **g, h** Plasma insulin levels and AUC of first-phase (0–5 min) and second-phase (5–120 min) insulin levels during hyperglycemic clamp in male choline diet-fed (18 weeks) *Fmo3*[+/+] and *Fmo3*[-/-] mice. *n* = 5 mice. **i–k** Immunofluorescence of insulin (green), glucagon (red) and DAPI (blue) in male *Fmo3*[+/+] and *Fmo3*[-/-] choline diet-fed (18 weeks) mice (**i**). This step was followed by measurements of % β-cell area (**j**) and % α cell area (**k**). *n* = 8 mice. Scale bar, 10 μm. Statistical significance was calculated (**b–h**, **j**) by two-sided Student's *t*-test. The data are presented as mean ± SEM. Mice: C57BL/6 J, choline diet feeding from 8 weeks old (**a–k**). Source data are provided as a Source Data file.

(Supplementary Fig. 4d). This evidence suggested that TMAO led to abnormal cytosolic calcium transients.

To further explore the mechanism, we performed RNA sequencing (RNA-seq) of MIN6 cells treated with TMAO (100 nM, 18 h)/vehicle with low or high glucose (Supplementary Fig. 4e). A total of 203 genes were upregulated and 382 genes were downregulated in the TMAO-treated cells under high-glucose conditions compared to the vehicle-treated cells under high-glucose conditions. Gene set enrichment analysis (GSEA) and Gene Ontology (GO) enrichment analysis showed

that TMAO significantly reduced insulin secretion, cytosolic calcium ion transport and calcium ion transmembrane transport and tended to decrease ER calcium ion homeostasis (Supplementary Fig. 4f). Since ER calcium ion homeostasis is crucial for maintaining cytosolic calcium ion levels[21,22] and TMAO increases resting cytosolic calcium levels (Fig. 4g), we analyzed ER calcium transport-related genes, including the sarcoendoplasmic reticulum (SR) calcium transport ATPase (Serca), which is responsible for calcium uptake to the endoplasmic reticulum (ER), and ryanodine receptor (Ryr) and inositol

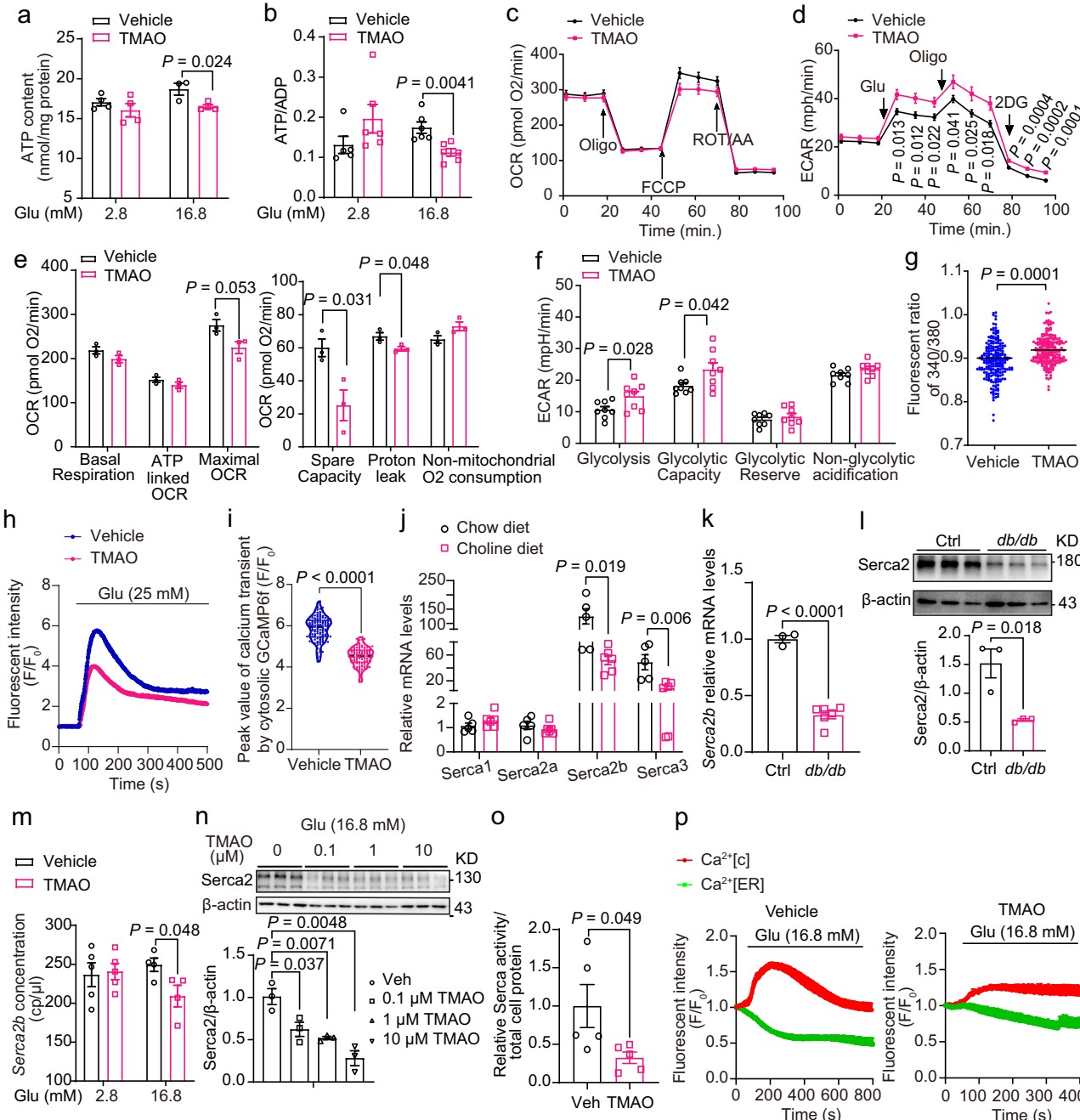

**Fig. 4 | TMAO inhibited glucose-stimulated mitochondrial respiration and cytosolic calcium transients. a** ATP content in MIN6 cells treated with or without TMAO (1 μM, 18 h). n = 3 (vehicle at 16.8 mM Glu), or 4 (others) biologically independent cell samples. **b** ATP/ADP in MIN6 cells treated with or without TMAO (100 nM, 6 h). n = 5 (vehicle at 2.8 mM Glu), or 6 (others) biologically independent cell samples. **c–f** OCR (**c, e**) and ECAR (**d, f**) of MIN6 cells treated with or without TMAO (100 nM, 18 h) by a Seahorse XFe96 analyzer. n = 3 (**c, e**), or 8 (**d, f**) biologically independent cell samples. **g** Resting state cytoplasmic calcium in MIN6 cells treated with (n = 222) or without (n = 223) TMAO (100 nM, 18 h). **h, i** Cytosolic calcium dynamics (**h**) and peak value of cytosolic calcium transient (**i**) stimulated by high glucose (25 mM) in β cells of Ins1-GCaMP6f mouse primary islets treated with (n = 22 in (**h**); n = 105 in (**i**)) or without (n = 46 in (**h**); n = 131 in (**i**)) TMAO (100 nM, 18 h). **j** Serca mRNA levels in islets isolated from male chow- and choline diet-fed (13 weeks) mice. n = 5 (chow), or 6 (choline) mice. **k, l** Serca2b mRNA and protein levels

in islets isolated from male control and *db/db* mice (12 weeks old). n = 3 (control), or 6 (*db/db*) mice (**k**); n = 3 mice (**l**). **m** *Serca2b* concentration in MIN6 cells treated with or without TMAO (1 μM, 18 h) by digital PCR. n = 5 (2.8 mM Glu), or 4 (16.8 mM Glu) biologically independent cell samples. **n, o** Serca2 protein levels and relative Serca activity/total cell protein in MIN6 cells treated with or without TMAO (100 nM, 1 μM or 10 μM, 18 h in (**n**); 100 nM, 18 h in (**o**)) under 16.8 mM glucose conditions. n = 3 (**n**), or 5 (**o**) biologically independent cell samples. **p** Glucose-stimulated cytoplasmic calcium and the release of ER calcium in MIN6 cells treated with or without TMAO (100 nM, 18 h). n = 3 biologically independent cell samples. Statistical significance was calculated (**a, b, d–g, i–o**) by two-sided Student's *t*-test. The data are presented as mean ± SEM. Oligo, oligomycin; FCCP, carbonyl cyanide 4-(trifluoromethoxy) phenylhydrazone; ROT, rotenone; AA, antimycin A. DG, 2-deoxy-D-glucose. Source data are provided as a Source Data file.

trisphosphate receptor (Itpr), which are responsible for calcium release from the ER to the cytosol. *Ryr and Itpr* were unchanged in the TMAO-treated group in response to 16.8 mM glucose (Supplementary Fig. 4g). However, *Serca2* and *Serca3* levels were decreased by TMAO upon 16.8 mM glucose stimulation (Supplementary Fig. 4g). In pancreatic β-cells, to maintain resting $Ca^{2+}$ levels, the $Ca^{2+}$ released during glucose stimulation must be extruded from the cytosol. Serca helps to lower cytosolic $Ca^{2+}$ by transporting $Ca^{2+}$ from the cytosol into the ER lumen, which requires ATP[21,23,24]. In the diabetic pancreas, Serca2b and Serca3 expression was reported to be reduced in β-cells, resulting in diminished insulin secretion[21,25]. Moreover, Serca2 overexpression resulted in increased GSIS[26]. Real-time PCR revealed that the mRNA levels of the pancreatic islet *Serca2b*, the most abundant subtype of Serca in the islets[26], and *Serca3* were significantly decreased in the choline diet-fed mice compared to the chow diet-fed mice (Fig. 4j). Consistently, *Serca2b* mRNA and protein levels in islets from *db/db* mice were significantly lower than those in the control group (Fig. 4k, l). Moreover, in MIN6 cells, TMAO (100 nM, 18 h) directly decreased the *Serca2b* and *Serca2a* mRNA levels with 16.8 mM glucose, with no change in the *Serca1* and *Serca3* mRNA levels (Supplementary Fig. 4h). Using digital PCR, which is more accurate for measuring mRNA content than qPCR, we confirmed that *Serca2b* levels were reduced by TMAO (Fig. 4m). In addition, the Serca2 protein abundance was reduced in MIN6 cells in a dose-dependent manner in response to 16.8 mM glucose (Fig. 4n and Supplementary Fig. 4i). Furthermore, Serca2 activity was inhibited by TMAO (Fig. 4o). Accordingly, under the high glucose (16.8 mM) stimulation, the release of ER calcium was attenuated in MIN6 cells treated by TMAO (Fig. 4p and Supplementary Fig. 4j).

## TMAO inhibited Serca2 expression through the NLRP3 inflammasome

Time course studies in choline diet-fed mice showed that Serca2 levels were decreased after 2 weeks of feeding, and pancreatic β cell dysfunction started to appear after 3 weeks of feeding (Supplementary Fig. 5a–c). Next, we investigated whether restoration of Serca2 function could rescue the effect of TMAO in vitro and in vivo. The Serca agonist [6]-gingerol[27] and the Serca2 agonist CDN1163[28] both reversed the inhibitory effect of TMAO on GSIS in MIN6 cells (Fig. 5a, b). Importantly, CDN1163 treatment improved glucose tolerance and GSIS in the choline diet-fed mice (Fig. 5c, d), with similar insulin sensitivity during the ITT and body weight or tissue weight between the groups (Supplementary Fig. 5d–f). Consistently, GSIS in cultured islets isolated from the CDN1163-treated mice showed higher insulin secretion under high-glucose conditions (Fig. 5e). Taken together, these data support the idea that Serca2 mediates the effect of TMAO on β-cell dysfunction.

Previous studies have indicated that TMAO promotes mitochondrial dysfunction and the activity of the NLRP3 inflammasome in macrophages[29] and that proinflammatory cytokines such as IL-1β lead to a loss of Serca2b expression in β-cells[26]. Next, we measured these factors in MIN6 cells. TMAO substantially enhanced mitochondrial reactive oxygen species (ROS) production in MIN6 cells and normal β cells from C57BL/6 J mice (Fig. 5f, g, and Supplementary Fig. 5g). Similarly, the ROS pathway was found to be enriched in GSEA in the TMAO-treated MIN6 cells in response to high glucose (Supplementary Fig. 5h), suggesting that mitochondrial function was impaired after TMAO treatment. Given that mitochondrial ROS activate the NLRP3 inflammasome[29], we next found that TMAO increased the levels of p-NF-κB, NLRP3, apoptosis-associated speck-like protein containing a caspase recruitment domain (ASC), cleaved caspase-1, cleaved IL-1β, and absent in melanoma 2 (AIM2), with lower Serca2 protein levels in MIN6 cells in response to high glucose (Fig. 5h–j), indicating that the NLRP3 inflammasome is activated by TMAO. Given that the proinflammatory cytokine IL-1β disturbs SERCA2 protein expression through Ser-273 phosphorylation of PPAR-γ in β-cells[26], we measured

the phosphorylation of the PPAR-γ protein at Ser-273 and found that it was increased by TMAO in response to 16.8 mM glucose (Fig. 5i, j). Moreover, MCC950, an NLRP3 inflammasome inhibitor, reversed the TMAO-induced loss of Serca2 in MIN6 cells (Fig. 5k, l). Collectively, our results indicated that TMAO impaired Serca2 through the NLRP3 inflammasome.

## TMAO diminished the β-cell proportion by promoting dedifferentiation and apoptosis and blocking transcriptional identity

The dramatic decrease in the β-cell percentage and the increase in the α-cell percentage in the choline diet-fed mice compared to the controls (Fig. 2m–o) indicated that the decreased β-cell percentage in choline diet-fed mice may be due to β-cell dedifferentiation and that TMAO might block β-cell maintenance. Therefore, we performed immunostaining for Sox9 and Neurogenin-3 (Ngn3), markers of failing or dedifferentiation of β-cells[9]. TMAO elevation in C57BL/6 J mice due to a choline diet increased the percentage of Sox9+/Ins+ cells (Fig. 6a, b) and Ngn3+/Ins+ cells (Supplementary Fig. 6a, b). Consistently, long-term TMAO (100 nM, 9 d) treatment obviously increased *Sox9* and *Ngn3* mRNA levels in MIN6 cells (Fig. 6c and Supplementary Fig. 6c). Similar to the effect of high glucose (35 mM glucose, 9 d) on β-cell dedifferentiation[30], long-term TMAO treatment increased the protein levels of Sox9, Ngn3 and another dedifferentiation marker, namely, chromograin A (ChgA), in MIN6 cells (Fig. 6d and Supplementary Fig. 6d). Consistently, mouse primary islets expressed higher Sox9, Ngn3 and ChgA protein levels after long-term TMAO treatment (Fig. 6e and Supplementary Fig. 6e). Together, our results suggested that long-term TMAO exposure directly led to β-cell dedifferentiation.

We next investigated whether TMAO affects the maintenance of β-cell transcriptional identity. We performed immunostaining for Pdx1 and Nkx6.1, two key β-cell transcription factors in islets[31]. The choline diet-fed mice exhibited decreased numbers of β-cells positive for Pdx1 (Fig. 6f, g) and Nkx6.1 (Supplementary Fig. 6f, g) compared to the controls. In addition, long-term TMAO treatment dramatically reduced the mRNA levels of the β-cell-specific markers *Ins1, Ins2, Iapp, Pdx1, Nkx6.1, Mafa, Ucn3 and Pcsk1* in MIN6 cells (Fig. 6h). Pdx1 protein levels were also decreased (Fig. 6i). These results suggested that long-term TMAO treatment reduced β-cell transcriptional identity.

Additionally, choline diet feeding increased the staining of cleaved caspase-3 (CC3), an apoptotic marker, in islets (Fig. 6j, k). Consistently, long-term TMAO (100 nM, 9 d) treatment increased CC3 and cleaved PARP (C-PARP) protein levels in MIN6 cells and mouse primary islets (Fig. 6l, m). These data indicated that long-term TMAO treatment directly led to β-cell loss by promoting β-cell apoptosis.

Serca2 reduction elicits ER stress[32], and ER stress leads to β-cell loss and dysfunction through β-cell dedifferentiation[33] and apoptosis[34]. Next, we investigated the effect of long-term TMAO treatment on ER stress and found that TMAO (100 nM, 9 d) increased ER stress-related protein levels, including those of p-PERK, p-eIF2α, Xbp1s and ATF6, in primary islets from C57BL/6 J mice and MIN6 cells (Fig. 6n, and Supplementary Fig. 6h).

## Reduction in TMAO levels by Fmo3 knockdown ameliorated insulin secretion deficits and β-cell loss in *db/db* mice

Because Fmo3 KO exerted a protective role against β-cell dysfunction, we examined the possibility of implementing Fmo3 as a therapeutic target to treat diabetes. We delivered an antisense oligonucleotide (ASO) targeting Fmo3[18] into *db/db* mice (Fig. 7a). This treatment resulted in an ~ 63% knockdown of *Fmo3* mRNA levels and a marked reduction in Fmo3 protein levels (Fig. 7b, c). Furthermore, *Fmo3*-ASO caused an ~64% plasma TMAO reduction and a 1.7-fold increase in Serca2 protein levels (Fig. 7d, e). Despite the comparable body weight, *Fmo3*-ASO resulted in decreased ratios of liver and epi-WAT weight to body weight and food intake (Supplementary Fig. 7a–c). The

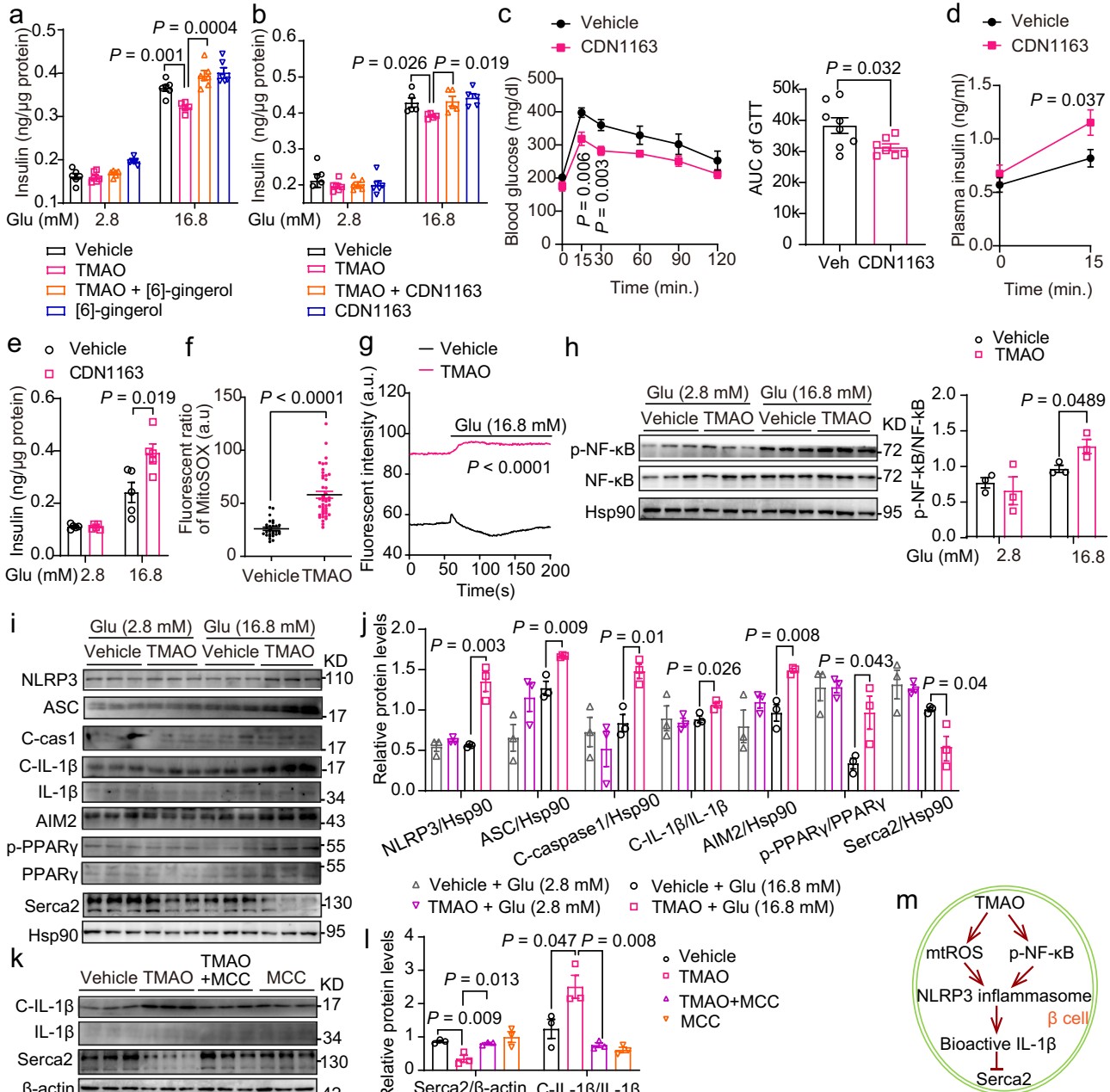

**Fig. 5 | TMAO signals through NLRP3 inflammasome activation to reduce Serca2 and GSIS. a** GSIS of MIN6 cells treated with TMAO (100 nM, 18 h) and the Serca agonist [6]-gingerol (0.08 μM, 10 min). $n = 6$ biologically independent cell samples. **b** GSIS of MIN6 cells treated with TMAO (100 nM, 18 h) and the Serca2b agonist CDN1163 (1 μM, 10 min). $n = 6$ (2.8 mM Glu), or 5 (16.8 mM Glu) biologically independent cell samples. **c** IPGTTs of vehicle- ($n = 8$ mice) and CDN1163-treated (6 d) male choline diet-fed (2 weeks) mice ($n = 7$ mice). Then, the AUC of the GTT was determined. **d** Plasma insulin levels at 0 and 15 min after intraperitoneal injection of glucose in vehicle- ($n = 7$ mice) and CDN1163-treated (19 d) male choline diet-fed (4 weeks) mice ($n = 5$ mice). **e** GSIS of primary islets from vehicle- and CDN1163-treated (40 d) male choline diet-fed (7 weeks) mice. $n = 5$ biologically independent islet samples. **f** Basal mitochondrial ROS measurement in MIN6 cells treated with ($n = 44$) or without ($n = 29$) TMAO (100 nM, 18 h). **g** Mitochondrial ROS measurement in MIN6

cells treated with ($n = 23$) or without ($n = 26$) TMAO (100 nM, 18 h) under 2.8 mM and 16.8 mM glucose conditions. **h–j** p-NF-κB, NF-κB, NLRP3, ASC, cleaved caspase-1 (C-cas1), cleaved IL-1β (C-IL-1β), p-PPARγ (Ser273 phosphorylation of PPAR-γ) and Serca2 protein levels in MIN6 cells treated with or without TMAO (10 μM, 18 h) under 2.8 mM and 16.8 mM glucose conditions. $n = 3$ biologically independent cell samples. **k, l** Cleaved IL-1β (C-IL-1β) and Serca2 protein levels in MIN6 cells treated with or without TMAO (10 μM, 18 h) or MCC (1 μM, 18 h) in response to 16.8 mM glucose. n = 3 biologically independent cell samples. **m** Diagrammatic sketch of the mechanism by which TMAO induces Serca2 loss. Statistical significance was calculated (**a–h, j, l**) by two-sided Student's *t*-test. The data are presented as mean ± SEM. MCC, MCC950 sodium; mtROS, mitochondrial reactive oxygen species. Source data are provided as a Source Data file.

intravenous glucose tolerance test (IVGTT) showed that *Fmo3*-ASO markedly improved glucose intolerance in *db/db* mice (Fig. 7f). As shown in Fig. 7g–j, *db/db* mice displayed impaired insulin secretion, as reflected by the decrease in insulin levels in response to glucose (1 min after glucose challenge), while *Fmo3*-ASO ameliorated insulin

secretion deficits. Similarly, the primary islets from the *Fmo3*-ASO-treated mice showed a dramatic improvement in insulin secretion by high glucose or KCl (Fig. 7k and Supplementary Fig. 7d).

Importantly, immunofluorescence studies also showed that insulin staining was increased by 1.4 times and glucagon staining was

decreased by 60% in the islets from the *Fmo3*-ASO mice, with a decrease in plasma glucagon levels and α-cell mass and increased β-cell mass in the *Fmo3*-ASO mice (Fig. 7l–n and Supplementary Fig. 7e–g). In parallel, TMAO reduction in *db/db* mice by *Fmo3*-ASO

virtually eliminated the presence of Sox9⁺ β-cells (Fig. 7o, p) and Ngn3⁺ β-cells (Supplementary Fig. 7h, i), indicating decreased β-cell dedifferentiation. Moreover, the *Fmo3* ASO-treated *db/db* mice displayed increased numbers of β-cells positive for Pdx1 (Fig. 7q, r) and Nkx6.1

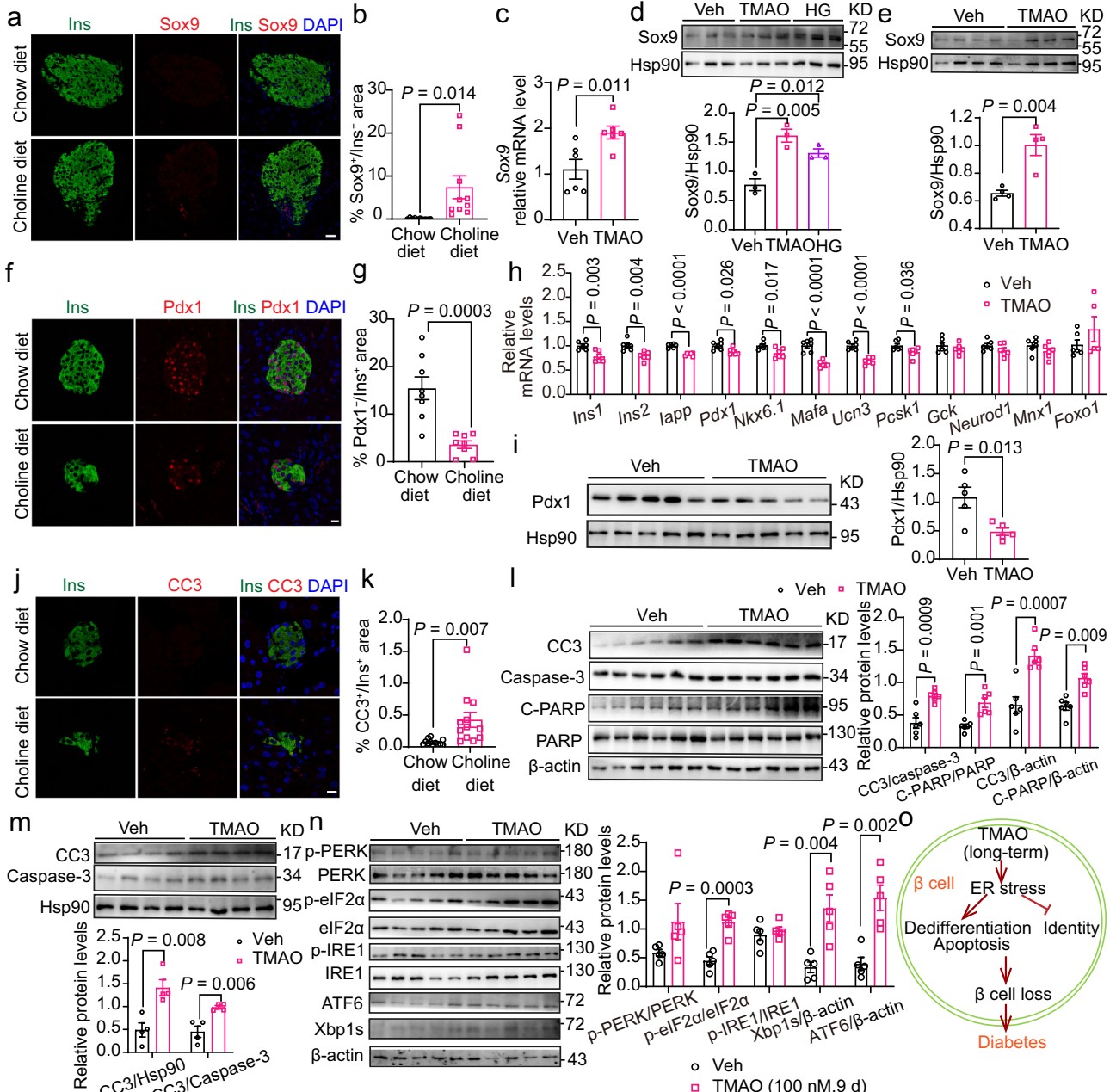

**Fig. 6 | TMAO promoted β-cell dedifferentiation and apoptosis and blocked β-cell transcriptional identity. a, b** Immunofluorescence of insulin (green), Sox9 (red) and DAPI (blue) in male chow- and choline diet-fed mice (**a**). Then, the % Sox9⁺/Ins⁺ area was measured (**b**). *n* = 10. Scale bar, 20 μm. **c** *Sox9* mRNA levels in MIN6 cells treated with and without TMAO (100 nM, 9 d). *n* = 6 biologically independent cell samples. **d** Sox9 protein levels in MIN6 cells treated with and without TMAO (100 nM, 9 d) as well as HG (35 mM glucose, 9 d). *n* = 3 biologically independent cell samples. **e** Sox9 protein levels in male mouse primary islets treated with and without TMAO (100 nM, 9 d). *n* = 4 biologically independent cell samples. **f, g** Immunofluorescence of insulin (green), Pdx1 (red) and DAPI (blue) in male chow- and choline diet-fed mice (**f**). Then, the % Pdx1⁺/Ins⁺ area was measured (**g**). *n* = 8. Scale bar, 10 μm. **h, i** β-cell markers mRNA levels in MIN6 cells and Pdx1 protein levels in male mouse primary islets treated with and without TMAO (100 nM, 9 d). *n* = 6 (**h**), or 5 (**i**) biologically independent cell sample.

**j, k** Immunofluorescence of insulin (green), CC3 (red) and DAPI (blue) in male chow- and choline diet-fed mice (**j**). This step was followed by measurements of % CC3⁺/ Ins⁺ area (**k**). *n* = 12. Scale bar, 10 μm. **l** Protein levels of the apoptosis markers CC3 and cleaved PARP (C-PARP) in MIN6 cells treated with and without TMAO (100 nM, 9 d). *n* = 6 biologically independent cell samples. **m** Protein levels of the apoptosis marker CC3 in male mouse primary islets treated with and without TMAO (100 nM, 9 d). *n* = 4 biologically independent cell samples. **n** ER stress-related protein levels in male mouse primary islets treated with or without TMAO (100 nM, 9 d). *n* = 5 biologically independent cell samples. **o** Diagrammatic sketch of the inhibitory effect of TMAO on β-cell maintenance. Statistical significance was calculated (**b–e, g–i, k–n**) by two-sided Student's *t*-test. The data are presented as mean ± SEM. Mice: C57BL/6 J, choline diet fed for 13 weeks from 8 weeks old (**a, b, f, g, j, k**); C57BL/6 J, 15 weeks old (**e, i, m, n**). Source data are provided as a Source Data file.

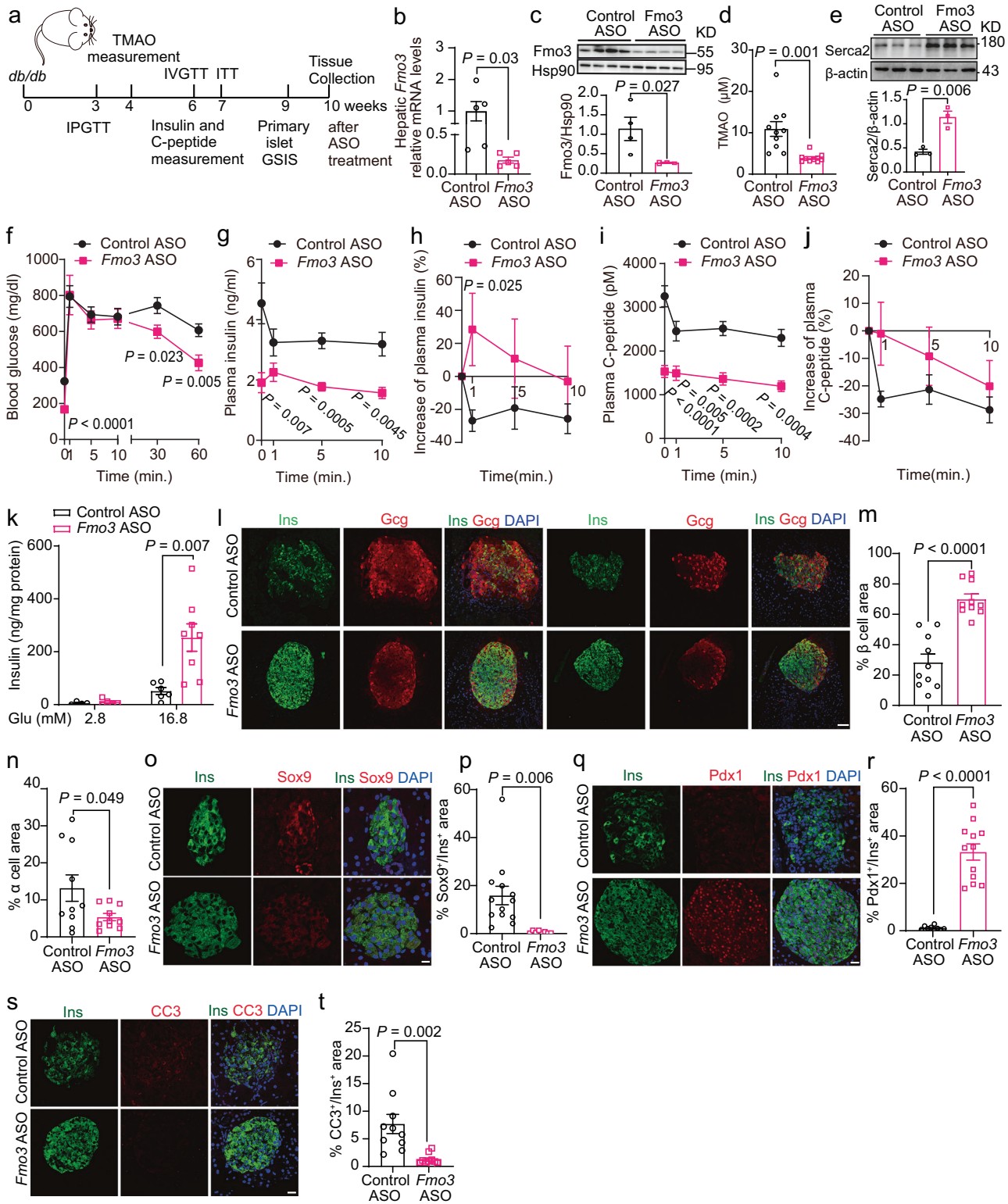

(Supplementary Fig. 7j, k) and a decreased number of CC3[+] β-cells (Fig. 7s–t) compared to the controls, showing maintained β-cell identity and inhibited β-cell apoptosis.

In addition, Fmo3 knockdown obviously decreased fasting insulin and C-peptide levels in *db/db* mice (Fig. 7g, i). Furthermore, IPGTT and ITT experiments indicated that Fmo3 knockdown improved fasting blood glucose, glucose intolerance, and insulin sensitivity in *db/db* mice (Supplementary Fig. 7l, m). The mRNA levels of the gluconeogenesis genes *G6pc*, *Pck1*, *Igfbp1*, *Fgf2* and *Foxo1* were reduced in the liver from the Fmo3 knockdown mice (Supplementary Fig. 7n). Together, these results showed that Fmo3 knockdown improved both β-cell function and insulin resistance in *db/db* mice.

## Discussion

Diabetes in mice and humans is associated with increasing TMAO levels, as we and others have shown[14,15] (Fig. 1a, i). In the livers of streptozotocin (STZ)-treated mice and *ob/ob* mice, *Fmo3* mRNA and protein expression were markedly induced[18]. Consistent with our

**Fig. 7 | Deficiency of Fmo3 improved β-cell function, β-cell loss and glucose homeostasis in *db/db* mice. a** Schematic diagram of tests during ASO injection. **b, c** Hepatic *Fmo3* mRNA and protein levels after 10 weeks of ASO treatment in male *db/db* mice. *n* = 5 (**b**), or 4 (**c**) mice. **d** Plasma TMAO concentration after 4 weeks of ASO treatment in male *db/db* mice. *n* = 10 mice. **e** Serca2 protein levels in primary islets from ASO-treated (6 weeks) male *db/db* mice. *n* = 3 mice. **f** IVGTTs after 6 weeks of ASO treatment in male *db/db* mice. *n* = 9 mice. **g, h** Plasma insulin (**g**) and increase in plasma insulin (%, **h**) during IVGTT. *n* = 7 (control ASO), or *n* = 6 (*Fmo3* ASO) mice. **i, j** C-peptide (**i**) and increase in C-peptide (%, **j**) during IVGTT. *n* = 7 mice. **k** GSIS in primary islets after 10 weeks of ASO treatment in male *db/db* mice. *n* = 4 (control ASO at 2.8 mM Glu), 5 (*Fmo3* ASO at 2.8 mM Glu), 6 (control ASO at 16.8 mM Glu), or 8 (*Fmo3* ASO at 16.8 mM Glu) biologically independent islet samples. **l–n** Immunofluorescence of insulin (green), glucagon (red) and DAPI (blue) in male control and *Fmo3* ASO-treated *db/db* mice (**l**). This step was followed by measurements of % β-cell area (**m**) and % α cell area (**n**). *n* = 10 mice. Scale bar, 50 μm. **o–t** Immunofluorescence for insulin (green; **o, q, s**), Sox9 (red; **o**), Pdx1 (red; **q**), CC3 (red; **s**) and DAPI (blue; **o, q, s**) in male control and *Fmo3* ASO-treated *db/db* mice. This step was followed by measurements of the % Sox9⁺/Ins⁺ area (**p**), % Pdx1⁺/Ins⁺ area (**r**), % CC3⁺/Ins⁺ area (**t**). *n* = 13 (control ASO), or 8 (*Fmo3* ASO) (**p**); *n* = 9 (control ASO), or 12 (*Fmo3* ASO) (**r**); *n* = 10 (**t**). Scale bar, 10 μm (**o**), or 20 μm (**q, s**). Statistical significance was calculated (**b–i, k, m, n, p, r, t**) by two-sided Student's *t*-test. The data are presented as mean ± SEM. Mice: *db/db*, ASO treatment (50 mg/kg body weight) from 6 weeks old (**a–t**). ASO, antisense oligonucleotides. Source data are provided as a Source Data file.

results (Fig. 1b, c), the levels of hepatic Fmo3, the TMAO-producing enzyme, were markedly elevated in *db/db* mice. In diabetes, intestinal dysbiosis is characterized by alterations in the abundance of many microbiota components, such as *Lachnospiraceae*, the abundance of which is enriched in T2D[35]. A significant elevation in the proportion of *Lachnospiraceae* is associated with higher plasma TMAO levels[36]. Thus, higher Fmo3 and gut microbiota levels could explain the elevation of the TMAO concentration in diabetes.

Increasing TMAO levels in C57BL/6 J mice by choline diet feeding impaired glucose tolerance, β-cell function and β-cell preservation, whereas it did not alter insulin sensitivity (Fig. 2) in these mice. Thus, in the choline diet-fed C57BL/6 J mice, glucose intolerance resulted from inadequate insulin secretion instead of insulin resistance. Similarly, reduction in TMAO levels by knockdown of Fmo3 in C57BL/6 J mice fed choline produced a state of improved GSIS and glucose intolerance (Fig. 3), with no effect on insulin sensitivity. These results demonstrated that in insulin-sensitive animal models, TMAO alone may affect only β-cell function but not insulin sensitivity. However, in insulin-resistant and diabetic *db/db* mice, TMAO knockdown by *Fmo3* ASO led to improvements in both β-cell function (Fig. 7) and insulin sensitivity (Supplementary Fig. 7). These results suggested that in animal models of disease, TMAO may also exacerbate insulin resistance in addition to β-cell dysfunction. In parallel with our results, Xiang Gao et al. reported that dietary TMAO increases the values of HOMA-IR, an index of insulin resistance, and exacerbates impaired glucose tolerance in insulin-resistant HFD-fed mice, with no effect on insulin-sensitive normal chow-fed mice[16].

In vivo, TMAO plasma levels show diurnal oscillations, which are relatively low during the light cycle and peak in the dark cycle[37]. In our study, most of the effects of TMAO on β-cells were observed at 100 nM or 1 μM for 18 h or even longer (9 d). In this case, the concentration is around the range observed in normal conditions, whereas TMAO treatment is always used. This finding may explain the harmful effect of TMAO at a low concentration.

In this study, we found that TMAO decreased Serca2 levels and that Serca2 agonists reversed the effect of TMAO on GSIS in vitro and in vivo, indicating that the major mechanism of TMAO action is probably the inhibition of cytosolic calcium transients through Serca2. Serca2 is a pump that transports calcium ions from the cytoplasm into the ER; this process is ATP-dependent[38,39], and a reduction in Serca2 protein expression blocks extracellular calcium influx and insulin secretion[26,40–44]. Several studies have reported a reduction in Serca2 levels in diabetic mice as well as human subjects. There was an 8- to 10-fold decrease in the Serca2 concentration in islets from *db/db* mice compared to that in the controls[26,45] and an approximately 70% decrease in the expression of Serca2b in islets from insulin receptor substrate-1 knockout (IRS-1 KO) mice, which exhibited defective GSIS, mild insulin resistance, and reduced insulin synthesis[46]. Similarly, in islets from subjects with T2D, the mRNA and protein levels of SERCA2b were reduced[26,47]. Impaired Serca function was reported to cause ER stress, which resulted in dedifferentiation and apoptosis in β-cells[32–34]. Therefore, long-term TMAO treatment-induced dedifferentiation and apoptosis may result from Serca reduction and disrupted calcium homeostasis.

Other researchers have noted that TMAO alleviates ER stress and could be beneficial. The dosage (100 mM, 2.78 mM with 0.25 μl/h infusion through subcutaneous osmotic minipumps) those authors used was far greater than that observed in vivo[48,49]. TMAO has been shown to correct protein folding defects indirectly by affecting hydrogen bonds within waters[50]. This molecule enhances the strength of water–water hydrogen bonds, which eventually stabilizes hydrogen bonds between amide groups in a protein and favors protein folding[51]. Of note, 54% of waters are within the spatially affected sphere of influence of TMAO in the 1 M TMAO simulation[51], which could explain why only high doses of TMAO have an inhibitory effect on ER stress. Furthermore, Emily S. Krueger et al. reported that TMAO protects INS-1 cell and rat islet function under diabetic glucolipotoxic conditions induced by 25 mM glucose and 0.5 mM palmitate[49]. Most likely, TMAO behaves differently under standard and harsh conditions, as the authors showed that TMAO had completely opposite effects on ER stress under glucolipotoxic and standard conditions.

The present study reveals the finding that the gut microbiota metabolite TMAO directly modulates β-cell function and maintenance in vitro and in vivo (Supplementary Fig. 7o). We reported a marked improvement in GSIS, β-cell proportion and glucose tolerance in Fmo3 knockdown mice. Furthermore, we demonstrated that TMAO directly decreases GSIS in human primary islets and that antisense Fmo3 improved β-cell function and glucose intolerance in *db/db* mice. Thus, inhibition of TMAO/Fmo3 could be an antidiabetic therapeutic strategy to alleviate β-cell dysfunction and glucose intolerance.

## Methods

### Study approval

All experiments using animals were performed following protocols approved by the Animal Experimentation Ethics Committee of the Chinese Academy of Medical Sciences, and all procedures were conducted following the guidelines of the Institutional Animal Care and Use Committees of the Chinese Academy of Medical Sciences. All animal procedures were consistent with the ARRIVE guidelines. Human primary islets were provided by Tianjin First Center Hospital (no. 2016N086KY). Human livers were provided by Wenzhou Medical University (no. 2016-246). Human serum was provided by Peking Union Medical College Hospital (no. ZS-1274). All protocols using human specimens were approved by the Institutional Review Board of the Chinese Academy of Medical Sciences and Peking Union Medical College. Informed consent was obtained from all subjects. The study conforms to the principles outlined in the Declaration of Helsinki. The clinical features of the patients are listed in Supplementary Tables 1–3 and Supplementary Data 1.

### Animal studies

Mice were maintained in a temperature- (21–23 °C) and humidity- (50–60%) controlled environment with a 12 h light/dark cycle (7 AM–7 PM), fed a standard laboratory chow diet (Beijing HFK Bioscience,

1035), administered water ad libitum, anesthetized with tribromoethanol and sacrificed at 2 PM in a fasted state. Plasma samples and tissues were stored at −80 °C prior to use. Male *db/db* mice and their lean, wild-type controls were purchased from GemPharmatech LLC. Male C57BL/6 J mice for chow (MolDiets, M18072501) and (MolDiets, M18072502) choline diet feeding were purchased from Beijing Vital River Laboratory Animal Technology Co., Ltd. *Fmo3*$^{-/-}$ mice were generated using Crispr/Cas9 (Biocytogen) as described previously[19]. Briefly, the gRNA was cloned and inserted into the Cas9 sgRNA vector, and in vitro transcription was performed (the gRNA sequence is listed in Supplementary Data 2). 16-week-old male Ins1-GCaMP6f mice, β-cell specific expressing GCaMP6f, were used for the islet β-cell calcium imaging experiments. GCaMP6f was a genetically encoded Ca$^{2+}$ indicator with green fluorescence. Ins1 cre-GCaMP6f mice were gifted by Prof. Liangyi Chen at Peking University, which were generated by crossbreeding GCaMP6f fl/fl mice (Jackson Laboratories, No.029626) and Ins1(Cre) mice[52] (Jackson Laboratories, No.026801). All diets were irradiated. The choline diet and the control chow diet were stored at −20 °C. Information of all diets was in Supplementary Table 4 and Supplementary Data 3. Diets were refreshed weekly, and body weight was measured weekly. For ASO experiments, four- to six-week-old male *db/db* mice were administered chemically modified ASOs (50 mg/kg body weight) by weekly intraperitoneal injection for 10 weeks. Control and Fmo3-specific chimeric 20-mer phosphorothioate oligonucleotides containing 2′O-methoxyethyl groups at positions 1 to 5 and 16 to 20 were diluted in 0.9% NaCl before injection[18]. For CDN1163 treatment experiments[28], 9-week-old male mice were fed a 1% choline diet and treated with vehicle (10% DMSO, 10% Tween 80 in 0.9% NaCl) or CDN1163 (50 mg/kg) intraperitoneally for 17 consecutive days (day 0 to day 16). Animals were monitored daily and humane endpoints for CO$_2$ euthanasia include maldevelopment, significant weight loss (>20% of total body weight), hunching, bleeding, infection, and/or reduced motility (inability to reach water and food).

## Cell culture

Cell lines were maintained at 37 °C in a 5% CO$_2$ cell culture incubator and passaged no more than 10 times. MIN6 cells were obtained from Dr. Tao Xu, a professor from the Institute of Biophysics, Chinese Academy of Sciences. αTC1-6 cells were purchased from ATCC (Catalog: CRL-2934). Cells were authenticated by the cell morphology and function. MIN6 cells were maintained in DMEM (25 mM glucose, Gibco) with 15% fetal bovine serum (FBS, Gibco), 50 μM β-mercaptoethanol (Sigma) and penicillin–streptomycin (Gibco). αTC1-6 cells were maintained in DMEM (1 g/l glucose, Gibco) with 10% fetal bovine serum (FBS, Gibco), 15 mM HEPES (Sigma), 0.1 mM nonessential amino acids (Gibco), 0.02% bovine serum albumin and penicillin–streptomycin (Gibco). Cell lines were routinely tested and verified to be mycoplasma negative using commercial mycoplasma detection kits (Londa, LT07-418).

Mouse primary islets were isolated as follows. After anesthetization with tribromoethanol, pancreases were inflated via the pancreatic duct with 3 ml of collagenase V in D-Hank's solution. Pancreas digestion was conducted in a 37 °C warm water bath for approximately 11 min. The samples were shaken violently for 15 s. To stop collagenase action, we washed the digest with ice-cold Hank's solution with 10% FBS, and all further steps were conducted on ice. Large tissue debris was removed by passing the digest through a 500 μm cell strainer. Then, islet cells were washed with Hank's solution once and transferred into islet culture medium (RPMI 1640 or low-glucose DMEM with 10% FBS, 1% penicillin and 1% streptomycin). Then, islet cells were separated using a stereomicroscope. After incubation overnight at 37 °C in a humidified incubator containing 5% CO$_2$, islets were hand-picked into new islet culture medium. Then, they were used for experiments.

The isolated islets were washed 2 times with PBS. They were then digested with 0.25% trypsin-EDTA (25300-062, Gibco) for 10–12 min at 37 °C followed by brief shaking. The digestion was stopped by RPMI 1640 culture medium with 10% FBS and penicillin–streptomycin and centrifuged at 96 g for 5 min. The cells were suspended in KRHB buffer with 2.8 mM glucose. Normal β cells were obtained through negative selection with magnetic activated cell sorting (MACS) using the α cell marker ACE2[53] (anti-ACE2-Biotin (Novus, cat. #NBP1-76614B, 1:10)), and the δ-cell marker CD24[54] (anti-CD24-Biotin (Miltenyi Biotec, clone M1/69, cat. #130-101-982, 1:10)) and identified by insulin staining. The cell suspension was plated on coverslips in a polyethylenimine-coated glass bottom dish (FD35-100, FluoroDish). The dishes were then kept for 2 h in a culture incubator at 37 °C and 5% CO$_2$ to allow cells to adhere. Additional culture medium was then added, and the cells were cultured for 12 h before the imaging experiments.

Human pancreases were obtained from organ procurement organizations and transported to the cell isolation facility, Tianjin First Center Hospital. The islet isolation procedures were performed as previously described[55]. All methods and protocols were carried out in accordance with guidelines and regulations by the Medical Ethics Committee of Tianjin First Center Hospital, and informed consent was obtained from every individual. All the experimental protocols, including any relevant details, were approved by the Medical Ethics Committee of Tianjin First Center Hospital. In general, the pancreas was distended with Serva NB1 enzymes (Serva, Germany) after trimming, and then, the tissue was digested using the modified Ricordi semiautomatic method for 8–10 min, washed and collected for further purification. After incubation in UW solution for 30 min, the digested tissue was purified using a UIC-UB gradient 42 in a Cobe 2991 cell separator (Cobe 2991, Cobe, USA). Finally, high-purity islets (more than 70%) were collected and cultured in CMRL culture media (Mediatech, USA) containing IGF (Cell Sciences, USA) and 20% human albumin (CSL Behring GmbH, Germany) at 37 °C in a 5% CO$_2$ incubator.

## Chemicals

TMAO (Sigma, Cat. 317594), D9-TMAO(Cambridge Isotopes,Cat. DML-4779-1), [6]-gingerol (Sigma, Cat. 345868), CDN1163 (Sigma, Cat. SML1682), MCC950 sodium (MCE, Cat. HY-12815A), Glucose (Sinopharm Chemical Reagent Co., Ltd. Cat. 10010592), GLP-1 (7-36) (MCE, Cat. HY-P0054), L-Arginine monohydrochloride (Sigma, Cat. A5131), Fatty acid free bovine serum albumin (Equitech-Bio, Inc.Cat. BAH66), Collagenase from Clostridium histolytium (Sigma, Cat. C9263), Human recombinant insulin (Lilly Cat. HI0219), Fura-2, AM (Invitrogen, Cat. F1221), mitoSOX Red, Molecular Probes (Invitrogen, M36008), phosphoenolpyruvate (Sigma, Cat. P7127), ATP disodium salt (Sigma, Cat. A3377), pyruvate kinase (Sigma, P1903-1KU), lactate dehydrogenase (Sigma, Cat. 59747), NADH (Sigma, Cat. N8129), digitonin (Sigma, Cat. D141), protease inhibitor cocktail (Sigma, Cat. P8340), PMSF (Roche, Cat. 10837091001), 4-Bromo A23187 (MCE, Cat. HY-N6694), Anti-Biotin MicroBeads (Miltenyi Biotec, Cat. 130-090-485).

## GTT, ITT, IVGTT and insulin measurement

Mice were fasted for 6 h for the GTT and ITT or overnight for the IVGTT. For the glucose tolerance test, glucose was injected intraperitoneally: a dose of 2 g/kg glucose was used in the experiments. For the ITT, insulin was injected intraperitoneally at a dose of 0.3 U/kg insulin in chow diet-fed and choline diet-fed mice or 1.2 U/kg insulin in *db/db* mice. For the IVGTT, glucose was injected via the tail vein, and a dose of 1 g/kg glucose was used in experiments. Blood glucose was measured with a glucometer (Roche) using whole blood from the tail. For the GTT and IVGTT, blood was also collected in heparinized capillary tubes (Thermo Fisher), centrifuged at 4000 × *g* for 10 min at 4 °C and used for measurement of insulin by mouse anti-insulin enzyme-linked immunosorbent assay (ALPCO).

## Hyperglycemic clamp

Insulin secretion in vivo was assessed by the one-step hyperglycemic clamp method[56]. The initial glucose load was 100 mg/kg. A 10% (for chow and choline diet mice) or 5% (for choline diet *Fmo3*[+/+] and *Fmo3*[−/−] mice) dextrose solution was infused through the jugular vein to clamp plasma glucose at 13.5–14.5 from 100–120 min and was adjusted based on glucose measurements (Roche). Plasma samples were collected from the tail at 0, 1, 5, 10, 30, 60, 90, and 120 min for measurements of insulin by enzyme-linked immunosorbent assays (ELISAs) (ALPCO).

## TMAO measurement

Liquid chromatography with tandem mass spectrometry (LC/MS/MS) was used for quantification of TMAO. Establishment of a standard curve: TMAO was dissolved in purified water and diluted to 0.2, 0.5, 1.5, 10, 20, 50, and 100 μM working solutions. Ten microliters of different concentrations of TMAO working solution was added to 40 μl of acetonitrile containing d9-TMAO (50 ng/ml, Cambridge Isotopes, DML-4779-1). The samples were vortexed, and 3 μl of supernatant was obtained for LC/MS/MS analysis. Plasma sample treatment: 10 μl plasma samples were added to 40 μl of acetonitrile containing d9-TMAO (50 ng/ml). The samples were vortexed and centrifuged 2 times (16,900 g, 5 min). Three microliters of supernatant was obtained for LC/MS/MS analysis. Super standard curve samples were diluted 100 times for determination. LC/MS/MS conditions: chromatographic column: Zorbax C18 (100 × 2.1 mm, 3.5 μm); column temperature 37 °C. Mobile phase: acetonitrile/water (containing 0.1% formic acid) gradient; the flow rate was 0.2 ml/min; TMAO was measured in positive ion mode monitoring at m/z 76.1 and m/z 85.2 for D9-TMAO.

## GSIS

MIN6 cells were cultured in 24-well plates ($8 \times 10^4$ cells/well) for 3 d, and mouse or human islet cells were cultured in dishes for 2 d. The cells were washed once with KRHB buffer (119 mM NaCl, 4.8 mM KCl, 2.5 mM CaCl$_2$, 1.2 mM MgSO$_4$, 1.2 mM KH$_2$PO$_4$, 25 mM NaHCO$_3$, 10 mM HEPES, 0.1% BSA) containing 2.8 mM glucose and fasted in KRHB buffer with 2.8 mM glucose for 1 h. Then, the cells were incubated in KRHB buffer with 2.8 or 16.8 mM glucose for 1 h. The supernatant was collected from each well, and insulin secretion was measured using an ELISA kit (ALPCO); values were normalized to the protein content of each well.

## Islet perifusion and time-lapse insulin secretion determination

After overnight culture, islets with similar sizes and statuses isolated from several mouse pancreatic tissues were randomly grouped into each group (30 islets per group). After preincubation in KRHB solution with 2.8 mM glucose for 1 h at 37 °C, islets were perfused with KRHB solution containing 2.8 mM glucose for 10 min and then stimulated with 16.8 mM glucose with TMAO by a self-designed device at a speed of 200 μl/min for 60 min. The solution was collected at 1 minute/point and mixed well to store at −20 °C for subsequent insulin concentration determination.

## HE staining

Tissues were fixed with 4% formalin for 24–48 h at room temperature with sufficient fixative to cover the tissues. The fixative volume was 5–10 times the tissue volume. The fixed tissues were trimmed to the appropriate size and shape and placed in embedding cassettes. The process for paraffin embedding schedule was as follows (total 16 h): 70% ethanol, two changes, 1 h each; 80% ethanol, one change, 1 h; 95% ethanol, one change, 1 h; 100% ethanol, three changes, 1.5 h each; xylene or xylene substitute (e.g., Clear Rite 3), three changes, 0.5 h each; paraffin wax (58–60 °C), two changes, 2 h each; and embedding tissues into paraffin blocks. Paraffin blocks were trimmed as necessary and cut at 3–10 μm (5 μm is commonly used). The paraffin ribbon was placed in a water bath at approximately 40–45 °C. The sections were mounted onto slides. The sections were allowed to air dry for 30 min and then baked in a 45–50 °C oven overnight. The baking temperatures were never higher than 50 °C for sections thicker than 25 μm. Otherwise, sections may crack, especially 25–50 μm thick sections, resulting in sections falling off slides during staining. The sections were deparaffinized in 2–3 changes of xylene for 10 min each. Three changes of xylene were used for sections thicker than 25 μm. The samples were hydrated in 2 changes of 100% ethanol for 5 min each and 95%, 80% and 70% ethanol for 5 min each. Then, the samples were rinsed in distilled water. The sections were stained in Harris hematoxylin solution for 5 min. The samples were washed in running tap water for 5 min, differentiated in 0.5% acid alcohol for 3–10 s, and washed in running tap water for 1 min. Then, blueing in 0.2% ammonia water or saturated lithium carbonate solution for 30 s to 1 min was performed. The samples were washed in running tap water for 5 min, rinsed in 95% alcohol, 10 dips, and counterstained in eosin-phloxine B solution (or eosin Y solution) for 30 s to 1 min. The samples were dehydrated through 95% alcohol and 2 changes of absolute alcohol for 5 min each. The samples were cleared in 2 changes of xylene, 5 minutes each, and mounted with xylene-based mounting medium.

## Immunofluorescence staining

The pancreases were fixed in 4% formalin and embedded in paraffin. The pancreatic sections were stained using insulin and glucagon antibodies and imaged using confocal microscopy (Zeiss). The pancreatic area and the insulin-, glucagon-, Sox9-, Pdx1-, CC3-, Ngn3-, and Nkx6.1-positive areas were measured using ImageJ software. The % β-cell area was calculated by dividing the total insulin-positive area by the total pancreatic islet area. The % α-cell area was calculated by dividing the total glucagon-positive area by the total pancreatic islet area. The β-cell mass was calculated by dividing the total insulin-positive area by the total pancreatic area and multiplied by the pancreatic weight. The α-cell mass was calculated by dividing the total glucagon-positive area by the total pancreatic area and multiplied by the pancreatic weight. The % Sox9$^+$/Ins$^+$, % Pdx1$^+$/Ins$^+$, % CC3$^+$/Ins$^+$, % Ngn3$^+$/Ins$^+$ and % Nkx6.1$^+$/Ins$^+$ areas were calculated by dividing the total Sox9-, Pdx1-, CC3-, Ngn3-, and Nkx6.1-positive areas by the total insulin-positive area, respectively. The antibody information is shown below: anti-insulin (R&D Systems, clone #182410, Cat #MAB1417, 1:200), anti-glucagon (Abcam, clone EP3070, #ab92517, 1:200), anti-Sox9 (Sigma Aldrich, #AB5535, 1:500), anti-Pdx1 (Abcam, clone EPR22002, cat. #ab219107, 1:1000), anti-Nkx6.1 (Abcam, clone EPR20405, cat. #ab221594, 1:100), anti-Neurogenin3 (Abcam, cat. #ab176124, 1:200), anti-cleaved caspase-3 (Asp175) (Cell Signaling Technology, clone 5A1E, cat. #9664, 1:400), Alexa Fluor 488 goat anti-rat (Invitrogen, cat. #A11006, 1:200), Alexa Fluor 546 goat anti-rabbit (Invitrogen, cat. #A11035, 1:200), Alexa Fluor 647 goat anti-mouse (Invitrogen, cat. #A32728, 1:200).

## Glucose uptake

MIN6 cells were cultured in 24-well plates ($8 \times 10^4$ cells/well) for 3 d. The cells were treated as indicated. The glucose uptake assay was performed following the Glucose Uptake-Glo Assay manufacturer's instructions. The medium was removed and washed with 500 μl of KRHB buffer. Cells were fasted in KRHB buffer for 30 min. A total of 250 μl of the prepared 1 mM 2-deoxy-D-glucose (2DG) or 25 mM 2DG was added to each well, shaken briefly, and incubated for 5 min at room temperature. Next, 125 μl of stop buffer was added and shaken briefly. Then, 75 μl of each sample was transferred to a 96-well plate. Next, 25 μl of neutralization buffer was added and briefly shaken. Then, 100 μl of 2-deoxy-D-glucose 6-phosphate (2DG6P) detection reagent was added and shaken briefly. The samples were incubated for 2 h at room temperature. Luminescence was recorded using a 0.3–1 s integration on a luminometer.

## ATP production

Sample Preparation: $1 \times 10^6$ MIN6 cells in 100 μl of ATP Assay Buffer were lysed. Next, 2–50 μl of sample was added to a 96-well plate. The volume was adjusted to 50 μl/well with ATP Assay Buffer. Standard Curve Preparation: Ten microliters of the ATP standard was diluted with 90 μl of $dH_2O$ to generate a 1 mM ATP standard and mixed well. Then, 0, 2, 4, 6, 8, and 10 μl was added to a series of wells, and the volume was adjusted to 50 μl/well with ATP Assay Buffer to generate the 0, 2, 4, 6, 8, and 10 nmol/well ATP standards. Reaction Mix: A sufficient amount of reagent for the number of samples and standards to be performed was used. For each well, 50 μl of Reaction Mix was used (45.8 μl of ATP Assay Buffer, 0.2 μl of ATP Probe, 2 μl of ATP Converter, 2 μl of Developer) and mixed well. Then, 50 μl of the reaction mix was added to each well containing the ATP standard and test samples. Measurement: The samples were mixed well. The samples were incubated at room temperature for 30 min in the dark. Fluorescence (Ex/Em = 535/587 nm) was measured in a microplate reader. The signals were stable for over 2 h.

## ATP/ADP ratio assay

Sample preparation: Cells were grown and incubated with vehicle or 100 nM TMAO (including a control culture without treatment). The culture medium was removed from the plate. Nucleotide release buffer (50 μl of buffer per $10^3–10^4$ cells) was added and incubated for 5 min at room temperature with gentle shaking. Assay procedure and detection: The reaction mix for each reaction was prepared as follows: 10 μl of ATP Monitoring Enzyme and 90 μl of Nucleotide Releasing Buffer. A sufficient amount of reagents for the number of assays (samples and induction control) to be performed was mixed. Then, 100 μl of the reaction mix was added to the control and sample wells, and the background luminescence was read (Data A). Then, 50 μl of cells ($10^3–10^4$ cells) treated with nucleotide releasing buffer was transferred into a luminometer plate. After approximately 2 min, the sample was read in a luminometer or luminescence capable plate reader (Data B). Next, 10× ADP-converting enzyme was diluted 10-fold with nucleotide releasing buffer. For measurement of ADP levels in the cells, the samples were read again (Data C), and then, 10 μl of 1× ADP Converting Enzyme was added. The samples were read again after approximately 2 min (Data D). The following equation was used: ADP/ATP ratio = [Data D - Data C] / [Data B - Data A]. The ATP/ADP ratio is the reciprocal of the ADP/ATP ratio.

## Whole-cell respiration and glycolysis analysis

Mitochondrial respiration was measured using the Seahorse XF24 Extracellular Flux Analyzer (Seahorse, Agilent, USA) according to the manufacturer's instructions. Briefly, MIN6 cells were seeded onto an XF24 microplate at 80,000 cells/well. The cellular OCR was monitored in buffered assay medium with 2 mM GlutaMAX, 2.5 mM sodium pyruvate, and 10 mM glucose (pH 7.4 at 37 °C), following the sequential addition of oligomycin (oligo, 1 μM), carbonyl cyanide 4-(trifluoromethoxy) phenylhydrazone (FCCP, 2 μM), rotenone (ROT, 1 μM) and antimycin A (AA, 1 μM). The maximal OCR was calculated by subtracting the OCR in the presence of rotenone and AA from that in the presence of FCCP. For glycolysis analysis, glucose (10 mM), oligomycin (1 μM) and 2-DG (42.5 mM) were applied.

## Mitochondrial ROS measurements

MitoSOX Red live-cell dye (M36008, Invitrogen™, Thermo Fisher Scientific) was used for mitochondrial ROS detection. For MitoSOX measurement, 2 μM indicator was loaded into MIN6 cells at 37 °C for 30 min in KRH solution and detected at the excitation wavelength (514 nm) and emission wavelength (580–740 nm) by a confocal microscope (LSM 880, Zeiss, Germany) with a 40×, 1.3 NA oil immersion objective.

## Calcium measurements in MIN6 cells and normal β cells of mouse primary islets

Calcium dyes were obtained from Thermo Fisher and dissolved in DMSO at a stock concentration of 1 mM. MIN6 cells were cultured in 35 mm glass bottom dishes (Cellvis, D35-14-1.5-N) and incubated for 15 min at 37 °C with Fura-2, AM (2 μM) in KRHB solution (119 mM NaCl, 4.8 mM KCl, 2.5 mM $CaCl_2$, 1.2 mM $MgSO_4$, 25 mM $NaHCO_3$, 1.2 mM $KH_2PO_4$, 2.8 mM glucose, 10 mM HEPES and 1% BSA) with 2.8 mM glucose, with or without 16.8 mM glucose stimulation. MIN6 cells were infected with a genetically encoded calcium indicator, GCaMP6f (Addgene, No.67564) expression adenovirus. After 18 h of infection, MIN6 cells were treated with TMAO (100 nM, 18 h) or vehicle. And cytoplasm calcium was measured after another 18 h. Mouse primary islets specific-expressed GCaMP6f in β cells were obtained by primary dissection and culture in 35 mm glass bottom dishes (Cellvis, D35-20-1.5-N, USA). The fluorescence intensity ratio of GCaMP6f was plotted against the dynamics of cytosolic calcium. The intracellular $Ca^{2+}$ concentration ($[Ca^{2+}]i$) was measured by confocal microscopy with Fura-2, a dual-wavelength ratiometric fluorescent calcium indicator. Fura-2 was excited by a laser at wavelengths of 340 and 380 nm, and $[Ca^{2+}]i$ was calculated from the ratio of the fluorescence intensity at 340 nm (F340) to that at 380 nm (F380). The fluorescence intensity ratio of Fura-2 was plotted against the dynamics of cytosolic calcium. A confocal microscope (Zeiss LSM 710) with a 40×, 1.3 NA oil-immersion objective was used for imaging. Frames of $512 \times 512$ pixels or $1024 \times 1024$ pixels were taken at a rate of 1 s/frame. The total number of frames and the length of imaging were dependent upon the experimental design. The digital image data were processed and analyzed by Zeiss Zen Blue software and ImageJ.

## RNA-seq

RNA quantification and qualification: RNA degradation and contamination were monitored on 1% agarose gels; RNA purity was assessed using the NanoPhotometer® spectrophotometer (IMPLEN, CA, USA); RNA integrity was assessed using the RNA Nano 6000 Assay Kit of the Agilent Bioanalyzer 2100 system (Agilent Technologies, CA, USA). Library preparation for lncRNA sequencing: A total amount of 1.5 μg RNA per sample was used as input material for the RNA sample preparations. Sequencing libraries were generated using the NEBNext® Ultra™ RNA Library Prep Kit for Illumina® (NEB, USA) following the manufacturer's recommendations, and index codes were added to attribute sequences to each sample. Briefly, mRNA was purified from total RNA using poly-T oligo-attached magnetic beads. Fragmentation was carried out using divalent cations under elevated temperature in NEBNext First Strand Synthesis Reaction Buffer (5×). First-strand cDNA was synthesized using random hexamer primers and M-MuLV Reverse Transcriptase (RNaseH-). Second-strand cDNA synthesis was subsequently performed using DNA Polymerase I and RNase H. Remaining overhangs were converted into blunt ends via exonuclease/polymerase activities. After adenylation of the 3' ends of DNA fragments, NEBNext adaptors with hairpin loop structures were ligated to prepare for hybridization. For selection of cDNA fragments with the right length, the library fragments were purified with an AMPure XP system (Beckman Coulter, Beverly, USA). Then, 3 μl of USER Enzyme (NEB, USA) was used with size-selected, adaptor-ligated cDNA at 37 °C for 15 min followed by incubation for 5 min at 95 °C before the PCR analysis. Then, PCR was performed with Phusion High-Fidelity DNA polymerase, Universal PCR primers and Index (X) Primer. Finally, the products were purified (AMPure XP system), and library quality was assessed on an Agilent Bioanalyzer 2100 system. Clustering and sequencing: The clustering of the index-coded samples was performed on a cBot Cluster Generation System using a HiSeq X-Ten/NovaseqS4 PE Cluster Kit (Illumina) according to the manufacturer's instructions. After cluster generation, the library preparations were sequenced on

an Illumina HiSeq X-Ten/NovaseqS4 platform, and 150 bp paired-end reads were generated.

## Real-time PCR analysis

Total RNA was extracted using TRIzol reagent (Life Technologies). cDNA was synthesized with 1 µg of total RNA according to the manufacturer's protocol (Applied Biosystems). The primer sequences are listed in Supplementary Data 2. The target cDNA was amplified by real-time PCR, and the results were normalized to the housekeeping gene 36B4. The value of the control group was set to 1.

## Western blot

Cells and tissues were lysed in RIPA buffer with protease inhibitors and phosphatase inhibitors for phosphorylated protein analysis. The protein concentration was measured using bicinchoninic acid (BCA). Equal amounts of protein were separated by SDS–PAGE and transferred to a PVDF membrane (Merck Millipore). Membranes were incubated in methanol for 1 min at room temperature, primary antibody (anti-Fmo3 (Abcam, clone EPR6968, cat. #Ab126711, 1:500), anti-Hsp90 (Proteintech, clone 3F11C1, cat. #60318-1-Ig, 1:5000), anti-ATP2A2/SERCA2 (Cell Signaling Technology (CST), clone D51B11, cat. #9580, 1:1000), anti-β Actin (Proteintech, clone 2D4H5, cat. #66009-I-Ig, 1:5000), anti-phospho-PERK (Thr980) (CST, clone 16F8, cat. #3179, 1:1000), anti-PERK (CST, clone C33E10, cat. #3192, 1:1000), anti-phospho eIF2α (Ser51) (CST, clone D9G8, cat. #3398, 1:1000), anti-eIF2α (CST, clone D7D3, cat. #5324, 1:1000), anti-phospho-IRE1 (Ser724) (Abcam, clone EPR5253, cat. #ab124945, 1:500), anti-IRE1 (CST, clone 14C10, cat. #3294, 1:1000), anti-XBP-1s (CST, clone E9V3E, cat. #40435, 1:1000), anti-ATF6 (Abcam, cat. #ab203119, 1:500), anti-Sox9 (Sigma Aldrich, cat. #AB5535, 1:500), anti-Pdx1 (Abcam, clone EPR22002, cat. #ab219207, 1:1000), anti-Nkx6.1 (Abcam, clone EPR20405, cat. #ab221549, 1:1000), anti-Neurogenin3 (Abcam, cat. #ab176124, 1:500), anti-ChromograinA (Abcam, clone EPR22537-248, cat. #ab254322, 1:1000), anti-cleaved caspase-3 (Asp175) (CST, clone 5A1E, cat. #9664, 1:1000), anti-caspase-3 (CST, clone D3R6Y, cat. #14220, 1:1000), anti-cleaved PARP (Asp214) (CST, clone 7C9, cat. #9548 1:1000), anti-PARP (CST, clone 46D11, cat. #9532, 1:1000), anti-NF-κB p65 (CST, clone D14E12, cat. #8242, 1:1000), anti-phospho-NF-κB p65 (Ser536) (CST, clone 93H1, cat. #3033, 1:1000), anti-NLRP3 (CST, clone D4D8T, cat. #15101, 1:1000), anti-ASC (CST, clone D2W8U, cat. #67824, 1:1000), anti-cleaved caspase-1 (CST, clone E2G2I, cat. #89332, 1:1000), anti-cleaved IL-1β (CST, clone E7V2A, cat. #63124, 1:1000), anti- IL-1β (CST, clone D6D6T, cat. #31202, 1:1000), anti-AIM2 (CST, cat. #63660, 1:1000), anti-phospho-PPAR-γ (ser273) (Bioss, cat. #bs-4888R, 1:500), anti-PPAR-γ (CST, clone C26H12, cat. #2435, 1:1000)) at 4 °C overnight, and then secondary antibody (HRP-conjugated goat anti-rabbit (Jackson ImmunoResearch, 111-035-003, 1:10,000) and HRP-conjugated goat anti-mouse (Jackson ImmunoResearch, 115-035-003, 1:10,000)) conjugated with horseradish peroxidase (Thermo Scientific) for 2 h at room temperature. The membranes were developed using chemiluminescent ECL reagents (Thermo Scientific).

## Serca activity assay

Ca$^{2+}$-dependent Serca activity was measured in whole lysates by a spectrophotometric assay using an enzyme-coupled system as previously described[57–59]. MIN6 cells were homogenized in homogenization buffer [250 mM sucrose, 5 mM HEPES pH 7.0, 25 µM digitonin, 1 mM PMSF, and complete protease inhibitor cocktail (Roche)]. The protein concentration was adjusted with homogenization buffer to 25 µg/mL and added to reaction buffer (100 mM KCl, 10 mM MgCl$_2$, 20 mM HEPES, pH 7.0, 10 mM phosphoenolpyruvate, 1 mM EGTA, 15 U/ml each of pyruvate kinase and lactate dehydrogenase, 0.5 mM NADH, 2 µM calcimycin A-23187, 5 µM free Ca$^{2+}$, final pH 7.0). Reactions were started by adding 5 mM ATP and incubated for 10 min at 37 °C, and

then, the decrease in absorbance at 340 nm was recorded for 10 min. Serca-independent Ca$^{2+}$-ATPase activity was measured in the presence of the Serca inhibitor thapsigargin (10 µM, BioVision 1558-1) and subtracted.

## Detection of ER calcium

Full length of G-CatchER+ sequences were cloned into the pAdeno vector (pAdeno-CMV-MCS) backbone and adenovirus packaging was produced by OBiO Technology Corp., Ltd (Shanghai, China). Cells were infected by adenovirus of G-CatchER+[60], and expresses over 36 h to ensure GCatChER protein localized in ER lumen, via a fused ER localization signal KDEL. Cells were loaded with 1 µM Rhod2, AM (Molecular Probe, R1245MP) for 10 minutes, 37 °C before time-lapse imaging.

## Statistics

No statistical method was used to determine animal's sample size. Sample size was chosen based on experience with the used experimental models in the field of cell biology and animal experiments. The number of samples and independent biological experimental repeats were indicated in the figure legends. Two-sided Student's $t$ tests are used in the statistics. Analyses were performed using Excel 2016 or GraphPad Prism 8.0 software with significance set to $P < 0.05$. In the figures, asterisks denote statistical significance ($*p < 0.05$; $**p < 0.01$; $***p < 0.001$). All quantitative data are expressed as mean ± SEM.

## Reporting summary

Further information on research design is available in the Nature Portfolio Reporting Summary linked to this article.

## Data availability

The RNA sequencing data generated in this study have been deposited in the Gene Expression Omnibus database under accession code GSE243083. All other data generated in this study are provided in the Article, Supplementary Information, or Source Data file. Source data are provided with this paper.

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

## Acknowledgements

We thank Dr. Tao Xu at the Institute of Biophysics, Chinese Academy of Sciences for providing the MIN6 cells. We thank Dr. Chu Wang at Peking University for advice. We thank Prof. Zhen Chai at Peking University for generously sharing an epi-microscope with 340 nm laser. We thank Prof. Heping Cheng at Peking University and Shiqiang Wang at Peking University for kindly providing support of G-Catch-ER+ plasmid. We thank Apex Biotechnologies (Suzhou) Co., Ltd. and STILLA TECHNOLOGIES, for help in technologies support the digital PCR data of this study. This work was supported by grants from the CAMS Innovation Fund for Medical Sciences (2021-I2M-1-016 to P.L., 2021-I2M-1-026 to Q.J., 2016-I2M-1-012 to L.K.), the Beijing Outstanding Young Scientist Program (BJJWZYJH01201910023028 to P.L.), the National Key R&D Program of China (2017YFA0205400 to P.L.), the National Natural Science Foundation of China (82104259 to Q.Z., 82104263 to Q.J. 81622010 to P.L., and 82304591 to Y.W.), the Chinese Academy of Medical Sciences (CAMS) Central Public-interest Scientific Institution Basal Research Fund (2017RC31009 and 2018PT35004), CAMS Innovation Fund for Medical Sciences (CIFMS) (2023-I2M-QJ-011 to L.K.), and the Special Research Fund for Central Universities, Peking Union Medical College (3332021041 to Q.Z., 3332022047 to Y.W.).

## Author contributions

L.K. performed most of the experiments and analyzed the data. Q.Z. performed whole-cell respiration and glycolysis analysis, analysis of calcium levels and mitochondrial ROS measurements. X.J. assisted with mouse primary islet isolation and qPCR experiments. Q.J. performed the OGTT, plasma insulin and C-peptide measurements of the control and *db/db* mice, and ATP/ADP assays and assisted with hyperglycemic clamp experiments. S.W. performed human islet isolation. X.J., Y.C. and Y.W. assisted with breeding mice and recording body weight and food intake. X.L., C.M. and Y.W. assisted with collecting tissues. S.H. performed an IPGTT of the *Fmo3⁻/⁻* mice fed a choline diet. L.S., J.H., L.Y. and L.Q. performed the TMAO measurements. X.P. and L.C. performed the islet perifusion assay. B.C. provided constructive advice. P.L. conceived the project and directed the research. L.K. and P.L. wrote the manuscript.

## Competing interests

The authors declare no competing interests.
