## [Peer Review File · Nature Communications]

Trimethylamine N-oxide impairs β -cell function and glucose toleranceREVIEWER COMMENTS

Reviewer #1 (Remarks to the Author):

This is a very interesting manuscript demonstrating the effects of TMAO in min6 cells, primary islets from db/db mice and primary human islets. The results demonstrate impairment in insulin secretion due to changes in SERCA2 expression. While the data is compelling, there are some concerns: Previous studies demonstrate no effect of TMAO under similar culture conditions in INS-1 cells and rat islets. It should be noted that INS-1 and min6 are cultured at different glucose conditions, with min6 cultured under exceptionally high glucose conditions.

Following are questions raised in the review:

Figure 1H-the effect of TMAO on GSIS is modest, however the effect on KCL induced secretion is much more striking. This is not discussed. Does this give a greater understanding of the effect of TMAO on the beta cell?

Figure 2C-please show area under the curve

Figure 2M-O-How many sections are being used from each animal? This is not defined. According to Chris Wright, a minimum of 5 sections per pancreas, sectioned throughout the entire pancreas, are needed to make a proper assessment of beta and alpha cell area.

Figure 3B-it is surprising that this crispr KO only shows a 50% reduction in protein. Do the original mice show only a 50% reduction?

Figure 3D-please show area under the curve

Figure 3I-K- How many sections are being used from each animal? This is not defined. According to Chris Wright, a minimum of 5 sections per pancreas, sectioned throughout the entire pancreas, are needed to make a proper assessment of beta and alpha cell area.

Figure 4G-Given that resting Ca²⁺ is higher with TMAO treatment, why is there no change in unstimulated insulin release? Shouldn't higher cytoplasmic Ca²⁺ result in greater basal insulin secretion?

Figure 5C-please show area under the curve

Figure 6A, F, I-the red image is very difficult to see, either bring up the gain or move to gray scale.

Figure 6G-please show western blotting for Pdx1

Figure 6K-it is odd that uncleaved caspase 3 does not decrease as cleaved increases. Is there greater expression of caspase 3?

Figure 7K, N, P, R-the red image is very difficult to see, either bring up the gain or move to gray scale.

Would db/db animals on a low choline diet have improved beta cell mass?

Do db/db islets have decreased serca2 protein levels? Does Fmo3 ASO KD increase serca2 levels in db/db islets? Does KCl induction normalize in db/db with Fmo3 ASO KD?

For sox9 and pdx1, what percent of the staining is nuclear?

How long is needed to see the phenotype?, to see SERCA2 decrease?

Reviewer #2 (Remarks to the Author):

In the manuscript "Trimethylamine N-oxide impairs beta-cell function and glucose tolerance", NCOMMS-23-01263A-Z, the authors suggest that Trimethylamine N-oxide (TMAO) may have a role in beta cell dysfunction and maintenance and thus that inhibition of TMAO could be a new way to treat T2DM.

Although this manuscript may be of potential interest there are a number of concerns that

need to be addressed.

Overall English should be checked. Why are both RNA and protein levels of Fmo3 increased in livers of db/db mice? Why are plasma TMAO levels decreased rather than increased in HFD-fed mice? ER stress was measured in MIN6-cells which are not normal beta cells.

Under conditions of elevated TMAO levels subsequent to dietary choline feeding why is there an increased glucagon staining and in this context what is the mechanistic explanation? What is the effect on alpha cell mass and function?

Under conditions of reduced TMAO levels subsequent to genetic deletion of Fmo3 what happens with alpha cell mass and function?

TMAO inhibited glucose-stimulated ATP production and cytosolic Ca²⁺ transients in MIN6 cells. Should also be shown for normal beta cells. With regard to ER Ca²⁺ handling, actual Ca²⁺ concentration needs to be measured within ER. Also release of Ca²⁺ from ER should be estimated.

Under the conditions where TMAO inhibited Serca2 expression through the NLRP3 inflammasome cytoplasmic Ca²⁺ concentration should be measured. Such measurements should also be performed in normal beta cells. When discussing Serca2, is it only the expression that is changed or is also the activity changed? The experiments where TMAO was shown to increase ROS production in MIN6 cells should also be extended to normal beta cells.

How should we consider the effect of TMAO on beta cell dedifferentiation, apoptosis and transcriptional identity? Which is the most important effect? Also, what happens with alpha cell mass under conditions of beta cell dedifferentiation?

Regarding reduction in TMAO levels by Fmo3 knockdown is this a global or liver specific knockdown?

Regarding the discussion on ER stress this part should be extended and more in-depth analyses should be done. Why are there different results and what does this actually mean?

In figure 1 l what is the n number. Big scatter of data. Really significant?

In figure 2 l it is difficult to see the arrow heads.

Why are figures 4 g and h displayed differently? Also, Ca²⁺ traces in h are not convincing. It is difficult to believe that there is no increase at all in Ca²⁺ subsequent to glucose stimulation in the TMAO treated group.

In figure 5k how should we consider reduced Serca2 protein levels in relation to Serca2 activity?

Response to the reviewers' comments

Reviewer #1 (Remarks to the Author):

This is a very interesting manuscript demonstrating the effects of TMAO in min6 cells, primary islets from db/db mice and primary human islets. The results demonstrate impairment in insulin secretion due to changes in SERCA2 expression. While the data is compelling, there are some concerns: Previous studies demonstrate no effect of TMAO under similar culture conditions in INS-1 cells and rat islets. It should be noted that INS-1 and min6 are cultured at different glucose conditions, with min6 cultured under exceptionally high glucose conditions.

R: We thank the reviewer for the encouraging words and comments. As the reviewer noted, MIN6 cells were cultured under high glucose conditions, which is quite different from INS-1 cells. Despite the different culture conditions, a previous study with INS-1 cells demonstrated a decreasing trend in GSIS in standard cell culture, and a dose of 80 μM TMAO significantly impaired GSIS in INS-1 cells and rat islets¹. This result is consistent with our findings.

Following are questions raised in the review:

Figure 1H-the effect of TMAO on GSIS is modest, however the effect on KCL induced secretion is much more striking. This is not discussed. Does this give a greater understanding of the effect of TMAO on the beta cell?

R: Insulin secretion can be induced by several factors, such as high glucose, KCl, and L-arginine. When the concentration of glucose increases, islet cell metabolism accelerates^{2, 3, 4}, and the ratio of cytosolic ATP to ADP is augmented^{4, 5}, which closes ATP-sensitive potassium channels (K_{ATP}) in the β -cell membrane^{6, 7} and leads to decreased potassium efflux, thus enabling depolarization of the plasma membrane⁸. The resulting depolarization is followed by the opening of several types of voltage-dependent calcium channels, which distinctly contribute to the acceleration of Ca^{2+} influx into the cell⁹. The ensuing increase in $[\text{Ca}^{2+}]_c$ ^{10, 11, 12} then activates an effector system that promotes exocytosis of insulin-containing granules. KCl and L-arginine cause membrane depolarization directly and subsequently induce insulin

secretion¹³. In our study, TMAO inhibited insulin secretion induced by high glucose, KCl, and arginine (Figures 1d, 1h and S1j in the original manuscript), indicating that the mechanism of TMAO is not upstream of membrane polarization. It has been reported that 30 mM KCl induces more insulin secretion than high glucose alone¹⁴. Consistently, our results showed that 30 mM KCl led to more robust insulin secretion than high glucose (Figure 1h in the original manuscript), which provided a larger window for TMAO to work on than glucose stimulation. Thus, the effect of TMAO on KCl-induced insulin secretion is more striking than that on GSIS.

Figure 2C-please show area under the curve

R: Thank you for the comment. We added data on the area under the curve (AUC) of the GTT (Figure 1). The AUC of the GTT in the choline diet-fed (5 weeks) mice was significantly higher than that in the chow diet-fed mice.

Figure 1. AUC of the IPGTT in choline diet-fed mice. Error bars represent the SEM, n=15 in each group. Mice: C57BL/6J, male, choline diet fed for 5 weeks from 8 weeks old. *p < 0.05.

Figure 2M-O-How many sections are being used from each animal? This is not defined. According to Chris Wright, a minimum of 5 sections per pancreas, sectioned throughout the entire pancreas, are needed to make a proper assessment of beta and alpha cell area.

R: We thank the reviewer for the comments. We used 5-6 sections per pancreas in Figures 2m-o. Below are the 5 sections in one pancreas (Figure 2), and the pictures in lane one are shown in Figure 2m of our original manuscript.

Figure 2. Five representative images of insulin (green), glucagon (red), and DAPI (blue) in paraffin-embedded pancreatic sections from male chow- and choline diet-fed (13 weeks) mice. Scale bar 20 μ m.

Figure 3B-it is surprising that this crispr KO only shows a 50% reduction in protein. Do the original mice show only a 50% reduction?

R: The KO mice in our manuscript showed a 60% reduction in Fmo3 protein (WT is approximately 1.4, and KO is approximately 0.5). We used the same strategy to knock out Fmo3 as described in the literature¹⁵. In the literature, the authors assessed the Fmo3 protein levels by western blotting and found that the Fmo3 band is almost completely gone in the liver of female ko mice. A previous report showed that Fmo3 levels in female mice are 100-fold higher than those in male mice¹⁶. In our study, we measured Fmo3 protein levels in male mice. Thus, the sex of the mice could be one reason to explain the relatively low reduction in our mice. The other reason would be the method of measurement. Generally, it is more accurate to use mRNA levels to reflect KO efficiency than protein levels, since truncated proteins can still respond to antibodies. The qPCR results showed an 80% reduction in Fmo3 in the KO group (Figure 3).

Figure 3. Hepatic *Fmo3* mRNA levels of *Fmo3*^{+/+} and *Fmo3*^{-/-} mice. Error bars represent the SEM, n=3 in each group. Mice: C57BL/6J, male, 8 weeks old. *p < 0.05.

Figure 3D-please show area under the curve

R: We added the AUC of the GTT in Figure 3d of the manuscript (Figure 4). The AUC of the GTT in *Fmo3*^{-/-} mice on a choline diet was significantly lower than that in *Fmo3*^{+/+} mice.

Figure 4. AUC of the IPGTT in *Fmo3* KO mice fed choline. Error bars represent the SEM, n=7-10 in each group. Mice: C57BL/6J, male, choline diet fed for 9 weeks from 8 weeks old. **p < 0.01.

Figure 3I-K- How many sections are being used from each animal? This is not defined. According to Chris Wright, a minimum of 5 sections per pancreas, sectioned throughout the entire pancreas, are needed to make a proper assessment of beta and alpha cell area.

R: We used 5 sections per pancreas in Figures 3i-k. Below are the 5 sections in one pancreas (Figure 5), and the pictures in lane one are shown in Figure 3i of our original manuscript.

Figure 5. Five representative images of insulin (green), glucagon (red), and DAPI (blue) in paraffin-embedded pancreatic sections from male choline diet-fed (18 weeks) *Fmo3*^{+/+} and *Fmo3*^{-/-} mice. Scale bar 10 μ m.

Figure 4G-Given that resting Ca^{2+} is higher with TMAO treatment, why is there no change in unstimulated insulin release? Shouldn't higher cytoplasmic Ca^{2+} result in greater basal insulin secretion?

R: Thank you for noting the effects of TMAO on resting calcium levels and basal insulin secretion in MIN6 cells. We found an increase in resting cytoplasmic calcium by TMAO under low glucose conditions (0.9 for vehicle, 0.92 for TMAO, $n=222-223$, Figure 4g in the original manuscript). A previous report showed that the first phase of glucose-induced insulin secretion requires a rapid and marked elevation of $[Ca^{2+}]_i$ in β cells¹⁷. The secretion efficiency of insulin granules in the first phase of secretion stimulated by high glucose was 6.7-20-fold higher than that at the resting phase per β cell per minute (2 granules vs. 0.1-0.3 granules/per β cell/per minute)⁶. GSIS requires a surge of intracellular calcium, whereas basal insulin secretion may not fully depend on resting calcium, and some studies suggest that basal hyperinsulinemia is driven by the synergistic actions of ROS and LC-CoA on exocytosis¹⁸. One example is the *db/db* mice, which showed higher resting calcium¹⁹, but the isolated islets from *db/db* mice did not show higher insulin secretion under low glucose conditions compared to controls (Figure 6).

Figure 6. GSIS in mouse primary islets from *db/db* mice. Error bars represent the SEM, n=4 in each group. Mice: male, 26 weeks old, *p < 0.05.

Figure 5C-please show area under the curve

R: We added the AUC of the GTT in Figure 5c of the original manuscript (Figure 7). The AUC of the GTT in CDN1163-treated choline diet-fed mice was significantly lower than that in the vehicle group.

Figure 7. AUC of the IPGTT in CDN1163-treated choline diet-fed mice. Error bars represent the SEM, n=7-8 in each group. Mice: C57BL/6J, male, choline diet fed for 2 weeks from 8 weeks old. *p < 0.05.

Figure 6A, F, I-the red image is very difficult to see, either bring up the gain or move to gray scale.

R: Thank you for the comment. We brought up the gain to make it easy to see (Figure 8).

Figure 8. TMAO promoted β -cell dedifferentiation and apoptosis and blocked β -cell transcriptional identity. **a** Immunofluorescence of insulin (green), Sox9 (red) and DAPI (blue) in chow- and choline diet-fed mice. Scale bar, 20 μ m. **b** Immunofluorescence of insulin (green), Pdx1 (red) and DAPI (blue) in chow- and choline diet-fed mice. Scale bar, 10 μ m. **c** Immunofluorescence of insulin (green), cleaved caspase-3 (CC3, red) and DAPI (blue) in chow- and choline diet-fed mice. Scale bar, 10 μ m. Mice: C57BL/6J, male, choline diet fed for 13 weeks from 8 weeks old.

Figure 6G-please show western blotting for Pdx1

R: Long-term TMAO (100 nM, 9 d) treatment reduced Pdx1 protein levels in mouse primary islets (Figure 9), which was consistent with the results in Figures 6g and h in the original manuscript.

Figure 9. Pdx1 protein levels in mouse primary islets treated with or without TMAO (100 nM, 9 d). Error bars represent the SEM, n=5 in each group. Mice: C57BL/6J, male, 15 weeks old. *p < 0.05.

Figure 6K-it is odd that uncleaved caspase 3 does not decrease as cleaved increases. Is there greater expression of caspase 3?

R: We repeated the experiments in MIN6 cells and mouse primary islets. Consistent with the finding in Figures 6k and l in the original manuscript, long-term TMAO (100 nM, 9 d) treatment significantly increased cleaved caspase-3 (CC3)/caspase-3 relative protein levels, with no decrease but a trend of increase in caspase-3/Hsp90 relative protein levels in both MIN6 cells and mouse primary islets (20-25%, Figure 10). Similarly, the mRNA levels of *caspase-3* showed an increasing trend after long-term TMAO treatment (100 nM, 9 d, Figure 10).

Figure 10. Effect of long-term TMAO treatment on apoptosis. **a** Protein levels of the apoptosis markers cleaved caspase-3 (CC3) and caspase-3 in MIN6 cells treated with and without TMAO (100 nM, 9 d). **b** Protein levels of the apoptosis marker cleaved caspase-3 (CC3) in mouse primary islets treated with and without TMAO (100 nM, 9 d). **c** *Caspase-3* mRNA levels in MIN6 cells treated with and without TMAO (100 nM,

9 d). Error bars represent the SEM, n=5-6 in each group. Mice: C57BL/6J, male, 15 weeks old. *p < 0.05, ***p < 0.001.

Figure 7K, N, P, R-the red image is very difficult to see, either bring up the gain or move to gray scale.

R: Thank you for the comment. We brought up the gain to make it easy to see (Figure 11).

Figure 11. Deficiency of Fmo3 improved β -cell loss in *db/db* mice. **a** Immunofluorescence staining for insulin (green), glucagon (red) and DAPI (blue) in the pancreas of control and Fmo3 ASO-treated *db/db* mice. Scale bar, 50 μ m. **b** Immunofluorescence staining for insulin (green), Sox9 (red) and DAPI (blue) in the pancreas of control and Fmo3 ASO-treated *db/db* mice. Scale bar, 10 μ m. **c** Immunofluorescence staining for insulin (green), Pdx1 (red) and DAPI (blue) in the pancreas of control and Fmo3 ASO-treated *db/db* mice. Scale bar, 20 μ m. **d** Immunofluorescence staining for insulin (green), cleaved caspase-3 (CC3, red) and DAPI (blue) in the pancreas of control and Fmo3 ASO-treated *db/db* mice. Scale bar, 20 μ m. Mice, male, 10 weeks of ASO treatment (50 mg/kg body weight) from 6 weeks old.

Would *db/db* animals on a low choline diet have improved beta cell mass?
 Do *db/db* islets have decreased serca2 protein levels? Does Fmo3 ASO KD increase serca2 levels in *db/db* islets? Does KCl induction normalize in *db/db* with Fmo3 ASO KD?

For sox9 and pdx1, what percent of the staining is nuclear?

R: It has been reported that in the livers of streptozotocin (STZ)-treated mice and *ob/ob* mice, *Fmo3* mRNA and protein expression are markedly induced²⁰. Consistent with our results (Figures 1b and c in the original manuscript), the mRNA and protein levels of hepatic *Fmo3*, the TMAO-producing enzyme, were dramatically elevated in *db/db* mice. Although *db/db* mice were fed a normal chow diet (0.125% choline), which has a much lower choline content than a choline diet (1% choline), they still produced much higher TMAO levels (Figure 1a in the original manuscript), indicating that in *db/db* mice, the higher expression of hepatic *Fmo3* led to an increase in plasma TMAO. Thus, if *db/db* mice are on a low-choline diet, we speculate that the TMAO levels will still be high and that the β cell mass will not be improved.

Yes, it has been reported that *Serca2* protein levels were decreased in islets from diabetic 20-week-old *db/db* mice compared with normoglycemic *db/+* littermate controls²¹. Our results showed that the *Serca2* protein levels were decreased in the islets of 12-week-old *db/db* mice compared to controls (Figure 12a). *Fmo3*-ASO treatment significantly increased *Serca2* protein levels in the islets of *db/db* mice compared to controls (Figure 12b). Consistently, in the islets of *Fmo3*-ASO-treated *db/db* mice, KCl induced greater insulin secretion compared to controls (Figure 12c).

In Figures 6a, f and 7n, p of the original manuscript, we showed that approximately 99% Sox9 and 100% Pdx1 staining was present in the nucleus.

Figure 12. Effect of *Fmo3* ASO on *Serca2* and insulin levels in islets of *db/db* mice. **a** *Serca2* protein levels in primary islets from Ctrl and 12-week-old *db/db* mice. **b** *Serca2* protein levels in primary islets from 6-week-old ASO-treated *db/db* mice. **c** KCl (30 mM) stimulated insulin secretion in primary islets after 5 weeks of ASO treatment

in *db/db* mice. Error bars represent the SEM, n=3 in each group (a-b), n=6 in each group (c). *p < 0.05, **p < 0.01. Mice, male, ASO treatment (50 mg/kg body weight) from 6 weeks old.

How long is needed to see the phenotype?, to see SERCA2 decrease?

R: Thank you for the question. We performed a time course experiment in C57BL/6J mice, which were fed a choline diet from 8 weeks old. Compared with those of the chow diet-fed mice, plasma TMAO levels started to increase from week 1; at that time, both Serca2 protein levels and insulin secretion after glucose challenge remained unchanged in choline diet-fed mice. Serca2 protein levels in the choline group were largely decreased from week 2 of the choline diet compared to the controls, with similar insulin secretion upon glucose between the choline and normal diet groups. After 3 weeks of choline diet feeding, insulin secretion upon glucose was significantly reduced in the choline group (Figure 13). Together, these results indicated that it takes two weeks for the choline diet to decrease Serca2 levels and 3 weeks to observe the phenotype.

Figure 13. The effect of a choline diet on TMAO, Serca2, and insulin secretion in C57BL/6J mice. **a** Plasma TMAO concentration in chow diet- and choline diet-fed (1, 2, 3 weeks) mice. **b** Serca2 protein levels in islets isolated from chow diet- and choline diet-fed (1, 2, 3 weeks) mice. **c** Plasma insulin levels after intraperitoneal injection of glucose for 0 and 10 min in chow diet- and choline diet-fed (1, 2, 3 weeks) mice. Error bars represent the SEM, $n=7-10$ in each group (a, c), $n=3$ in each group (b). Mice: C57BL/6J, male, choline diet fed for 1, 2, and 3 weeks from 8 weeks old. * $p < 0.05$, ** $p < 0.01$.

Reviewer #2 (Remarks to the Author):

In the manuscript "Trimethylamine N-oxide impairs beta-cell function and glucose tolerance", NCOMMS-23-01263A-Z, the authors suggest that

Trimethylamine N-oxide (TMAO) may have a role in beta cell dysfunction and maintenance and thus that inhibition of TMAO could be a new way to treat T2DM.

Although this manuscript may be of potential interest there are a number of concerns that need to be addressed.

Overall English should be checked. Why are both RNA and protein levels of *Fmo3* increased in livers of *db/db* mice? Why are plasma TMAO levels decreased rather than increased in HFD-fed mice? ER stress was measured in MIN6-cells which are not normal beta cells.

R: Thank you for the reviewer's comments. We have used a professional English editing company to review the language.

Previous studies showed that testosterone is responsible for the lower hepatic *Fmo3* expression in males, and estrogen induces *Fmo3* expression in females²². Testosterone levels in *db/db* mice were lower than those in control *db/+* mice²³. The reduced testosterone levels might be the reason for the increased *Fmo3* expression in the livers of *db/db* mice.

In addition to *Fmo3*, plasma TMAO levels are correlated with gut microbiota. Compared with those of the normal chow diet-fed mice, hepatic *Fmo3* mRNA levels were unchanged in the HFD-fed (19 weeks) mice (Figure S1f of the original manuscript). *Firmicutes* and *Bacteroidetes* are dominant bacteria in the intestinal flora of mice at the phylum level. It has been reported that TMAO levels are positively correlated with *Bacteroidetes* and negatively correlated with *Firmicutes*²⁴. Moreover, a high-fat diet significantly increases the relative abundance of *Firmicutes* and decreases that of *Bacteroidetes*²⁵. These studies indicate that gut microbiota dysbiosis leads to decreased plasma TMAO levels in HFD-fed mice.

A long-term low dose of TMAO is needed to induce ER stress. As shown in Figure 6m (in the original manuscript) in MIN6 cells, 9 days of TMAO (100 nM) increased the markers of ER stress. As suggested, we attempted to repeat these experiments in normal β cells. Unfortunately, the isolated normal β cells did not survive for 9 days. Normal β cells are very fragile, and culturing normal β cells is challenging in the field of diabetes.

We then repeated these experiments in isolated islets from C57BL/6J mice and found that long-term TMAO treatment (100 nM, 9 d) increased ER stress-related protein levels, including those of p-PERK, p-eIF2 α , Xbp1s and ATF6, in primary islets (Figure 14), which is consistent with the results in MIN6 cells (Figure 6m in the original manuscript).

Figure 14. ER stress-related protein levels in mouse primary islets treated with or without TMAO (100 nM, 9 d). Error bars represent the SEM, n=5 in each group. Mice: C57BL/6J, male, 15 weeks old. ***p < 0.001.

Under conditions of elevated TMAO levels subsequent to dietary choline feeding why is there an increased glucagon staining and in this context what is the mechanistic explanation? What is the effect on alpha cell mass and function?

R: Thank you for the comment. Dietary choline feeding led to β -cell dedifferentiation with increased numbers of β -cells positive for Sox9 (Figures 6a-b of the original manuscript). Sox9 is a pluripotent marker, and studies have shown that pluripotent β cells can be converted to α cells^{26, 27}. Choline diet-fed mice exhibited decreased numbers of β -cells positive for Pdx1 (Figures 6f and g of the original manuscript). Furthermore, it has been reported that adult β cells rapidly acquire ultrastructural and physiological features of α cells after Pdx1 deletion²⁸. In addition, long-term TMAO treatment dramatically reduced the mRNA levels of the β -cell-specific markers *Pdx1*, *Ins1*, *Ins2*, *Iapp*, *Nkx6.1*, *Mafa*, *Ucn3* and *Pcsk1* in MIN6 cells (Figure 6h in the original manuscript). Thus, β -cell dedifferentiation eventually led to increased glucagon staining in our study. The mechanisms could still be ER stress, since it has been reported that ER stress leads to β -cell loss and dysfunction through β -cell dedifferentiation²⁹ and apoptosis³⁰.

We calculated α -cell mass and found the choline diet increased α -cell mass (Figure 15a). Consistently, there was more glucagon in the plasma of the choline diet-fed mice than in the controls (Figure 15b). To determine whether TMAO has a direct effect on α cells, we cultured the α cell line α TC1-6 and measured glucagon levels under low glucose conditions. The results showed that TMAO had no effect on glucagon secretion under low glucose condition (Figure 15b). Taken together, these data indicate that TMAO does not have a direct effect on α cell function and that β -cell dedifferentiation might account for the increase in α -cell mass and plasma glucagon levels.

Figure 15. Dietary choline feeding increased fasting plasma glucagon levels. a Quantification of α -cell mass in chow- and choline diet-fed (13 weeks) mice. **b** Plasma glucagon levels of chow- and choline diet-fed (6 weeks) mice. **c** Glucagon secretion under low glucose conditions in α cell line α TC1-6 treated with or without TMAO (100 nM, 18 h). Error bars represent the SEM, n=6 in each group (a), n=10 in each group (b), n=4 in each group (c). Mice: C57BL/6J, male, choline diet feeding from 8 weeks old. **p < 0.01.

Under conditions of reduced TMAO levels subsequent to genetic deletion of Fmo3 what happens with alpha cell mass and function?

R: The treatment of *db/db* mice with Fmo3 ASO (10 weeks) resulted in a reduction in plasma TMAO and glucagon staining in the pancreas (Figure 7k in the original manuscript). After calculation, α cell-mass in the pancreas was decreased by 50% in the Fmo3 ASO group (Figure 16a), indicating that Fmo3 ASO reduced α -cell mass in *db/db* mice. Consistently, we found a decrease in plasma glucagon in the Fmo3-ASO group (Figure 16).

Figure 16. Fmo3 knockdown by ASO treatment decreased α -cell mass and fasting plasma glucagon levels in *db/db* mice. **a** Quantification of α -cell mass in control and Fmo3 ASO-treated (10 weeks) *db/db* mice. **b** Plasma glucagon levels of control and Fmo3 ASO-treated (6 weeks) *db/db* mice. Error bars represent the SEM, n=6-10 in each group. Mice: male, ASO treatment (50 mg/kg body weight) from 6 weeks old. * $p < 0.05$.

TMAO inhibited glucose-stimulated ATP production and cytosolic Ca^{2+} transients in MIN6 cells. Should also be shown for normal beta cells. With regard to ER Ca^{2+} handling, actual Ca^{2+} concentration needs to be measured within ER. Also release of Ca^{2+} from ER should be estimated.

R: As suggested, we isolated normal β cells from C57BL/6J mice and treated them with TMAO (100 nM, 18 h) or vehicle. Unfortunately, culturing normal β cells is pretty challenging, and there were not enough β cells for the measurement of ATP content. We can obtain about 50 thousand alive normal β cells from one mouse. After overnight of culture, only 2 thousand β cells survived. At least 100 mice would be sacrificed to obtain enough β cells for this assay, which is really a waste of mice. We then isolated islets from C57BL/6J mice and treated them with TMAO (100 nM, 18 h), followed by the measurement of ATP content. As shown in Figure 17, TMAO decreased ATP production by glucose, which is consistent with the finding in MIN6 cells (Figure 4a in the original manuscript).

To measure the effect of TMAO on cytosolic calcium in normal β cells, we first fed C57BL/6J mice with choline for 9 weeks, the normal β cells were then isolated. After overnight culture, the normal β cells attached to the bottom, followed by high glucose stimulation and calcium measurement with Fluo4. The results only showed a trend of decrease in cytosolic calcium levels in the normal β cells from choline-treated mice compared to those in β cells from chow-fed mice (Figure 17b). The individual cell trace

was listed in Figures 17c and d. During the overnight culture, more than 95% of the normal β cells died, and some of the survivors died during the experiment with measurement of calcium probably due to the laser irradiation. So we don't think it's appropriate to make any conclusion with Figure 17b.

We tried hard to measure ER calcium in normal β cells. Unfortunately, the normal β cells infected with the virus expressed ER located calcium sensor G-CatchER⁺³¹ did not respond to high glucose. To monitor the effect of TMAO on Ca²⁺ concentrations within the ER lumen, we then expressed G-CatchER⁺³¹ in MIN6 cells. Before monitoring calcium of endoplasmic reticulum (ER) via G-Catch-ER⁺ indicator, we investigated the spatial localization of G-Catch-ER⁺, and observed perfect co-localization of G-Catch-ER⁺ and ER marker Sec61b (Figure 17e), which was the central component of the protein translocation apparatus of the ER membrane. This result confirmed that G-Catch-ER⁺ could localize in the ER of MIN6 cells. Meanwhile, we applied fluorescent dye Rhod-4 to simultaneously detect cytosolic calcium in the same cells. The results showed that the decreasing amplitude of calcium within the ER lumen (Ca²⁺[ER]) was much less, accompanied by lower cytosolic calcium transient triggered by high glucose, in the TMAO group than in the controls. It demonstrated that under the high glucose (16.8 mM) stimulation, the release of ER calcium was attenuated in MIN6 cells treated by TMAO, which may affect cytosolic calcium transient and insulin secretion (Figure 17f).

Figure 17. TMAO inhibited glucose-stimulated ATP production, cytosolic Ca^{2+} transients and the release of ER calcium. **a** ATP content in mouse primary islets treated with or without TMAO (100 nM, 18 h). **b** Basal and glucose-stimulated cytoplasmic calcium in normal β cells from chow- and choline diet-fed (9 weeks) mice. **c** The individual cells traces of the chow diet group. **d** The individual cells traces of the choline diet group. **e** Spatial localization of ER calcium indicator G-Catch-ER+ and ER marker Sec61 β in MIN6 cells. Scale bar: 10 μm . **f** The effect of TMAO (100 nM, 18 h) on ER calcium in MIN6 cells. Time-lapse imaging was applied to simultaneously monitor cytosolic calcium ($\text{Ca}^{2+}[\text{c}]$) and ER calcium ($\text{Ca}^{2+}[\text{ER}]$), cytosolic Ca^{2+} was detected by Rhod-2, and ER Ca^{2+} via G-CatchER+. Error bars represent the SEM, $n=6$ in each group (a), $n=3$ in each group (b, f). Mice: C57BL/6J, male, 15 weeks old (a); C57BL/6J, male, choline diet fed for 9 weeks from 8 weeks old (b-d). * $p < 0.05$, ** $p < 0.01$.

Under the conditions where TMAO inhibited Serca2 expression through the NLRP3 inflammasome cytoplasmic Ca^{2+} concentration should be measured. Such measurements should also be performed in normal beta cells. When discussing Serca2, is it only the expression that is changed or is also the activity changed? The experiments where TMAO was shown to

increase ROS production in MIN6 cells should also be extended to normal beta cells.

R: Regarding cytoplasmic calcium in normal β cells, please see the response to Figure 17.

We found that TMAO decreased Serca2 protein levels in MIN6 cells by western blotting (Figure 18a). Next, we measured Serca2 activity by a spectrophotometric assay using an enzyme-coupled system as previously described^{32, 33, 34} and found that TMAO greatly inhibited Serca2 activity (Figure 18b). When the activity was normalized by Serca2 protein levels, there was no decrease in Serca2 activity in the TMAO-treated group (Figure 18c). These results indicate that the reduction in Serca2 activity was mainly due to the decrease in Serca2 protein levels.

To measure the effect of TMAO on mitochondrial ROS in normal β cells, we fed C57BL/6J mice choline. After 7 weeks of feeding, the normal β cells were isolated, followed by mitochondrial ROS detection by MitoSOX Red live-cell dye. The results showed that in normal β cells from choline-treated mice, the mitochondrial ROS levels were greatly increased compared to those in β cells from chow-fed mice (Figure 18d), which is consistent with the finding in MIN6 cells (Figure 5g in the original manuscript).

Figure 18. Effect of TMAO on Serca activity and mitochondrial ROS. a-c Serca2 protein levels (a), relative Serca activity/total cell protein (b) and relative Serca activity/Serca2 protein (c) in MIN6 cells treated with or without 18 h of TMAO (100 nM) under 16.8 mM glucose conditions. d Mitochondrial ROS in normal β cells from chow- and choline diet-fed mice. Error bars represent the SEM, n=5 in each group (a-c), n=9 in the chow diet group and n=6 in the choline diet group (d). Mice: C57BL/6J, male, choline diet fed for 7 weeks from 8 weeks old. *p < 0.05, ***p < 0.001.

How should we consider the effect of TMAO on beta cell dedifferentiation,

apoptosis and transcriptional identity? Which is the most important effect? Also, what happens with alpha cell mass under conditions of beta cell dedifferentiation?

R: In our study, the effect of TMAO on β -cells occurred through Serca2 reduction and an imbalance in ER calcium homeostasis, which elicits ER stress. Since ER stress leads to β -cell loss and dysfunction through β -cell dedifferentiation²⁹ and apoptosis³⁰, we think that the ER stress induced by TMAO directly led to β -cell dedifferentiation and apoptosis, which subsequently led to the loss of β -cell transcriptional identity. Therefore, ER stress was the most important effect of TMAO in our study. After choline diet feeding, β -cell dedifferentiation displayed more significant changes than apoptosis and transcriptional identity (50 vs. 5, Figures 6a, b, f, g, i, j of the original manuscript). This finding suggested that β -cell dedifferentiation may be the most important effect on TMAO-impaired β -cell function.

Under conditions of choline diet feeding in C57BL/6J mice, α -cell mass was increased by 4-fold, with a 50-fold increase in β -cell dedifferentiation, suggesting elevated α -cell mass in choline diet-fed mice (Figure 15a).

Regarding reduction in TMAO levels by Fmo3 knockdown is this a global or liver specific knockdown?

R: The antisense oligonucleotides (ASOs) of Fmo3 lead to global knockdown in *db/db* mice. Therefore, Fmo3 levels were decreased in multiple tissues, such as the liver (Figures 7b-c in the original manuscript), kidney, quadricep, and epi-WAT (Figure 19).

Figure 19. Fmo3 knockdown using antisense oligonucleotides (ASOs) is a global knockdown. Fmo3 ASO knockdown *Fmo3* mRNA levels of quadricep, kidney and epi-WAT in *db/db* mice. Error bars represent the SEM, n=4-5 in each group. Mice: male,

ASO treatment (50 mg/kg body weight, 10 weeks) from 6 weeks old. * $p < 0.05$, ** $p < 0.01$.

Regarding the discussion on ER stress this part should be extended and more in-depth analyses should be done. Why are there different results and what does this actually mean?

R: Thank you for the suggestions. Here, we showed that TMAO at low concentrations (100 nM, 1 μ M) promoted ER stress and inflammation. This result is opposite to the finding with super high levels of TMAO (100 mM)³⁵ or under glucolipotoxic conditions (25 mM glucose and 0.5 mM palmitate³⁶). When a high dose of TMAO shows a beneficial effect on ER stress, the cells or tissues are usually already under severe ER stress conditions, indicating that proteins are unfolded or misfolded. Protein stability is the result of a balance between the intramolecular interactions of protein functional groups and their interactions with the solvent environment³⁷. TMAO has been shown to correct protein folding defects indirectly by affecting hydrogen bonds within waters³⁸. When a protein folds, hydrogen bonds between amide groups form in the protein interior, whereas when a protein unfolds, hydrogen bonds between amide groups in the protein are broken and replaced with hydrogen bonds with water. TMAO could enhance the strength of water–water hydrogen bonds, which eventually stabilizes hydrogen bonds between amide groups in a protein and favors protein folding³⁹. Of note, 54% of waters are within the spatially affected sphere of influence of TMAO in the 1 M TMAO simulation and 86% at 2 M TMAO, suggesting that a high amount of TMAO is needed to increase water–water hydrogen bonds³⁹. This finding could also explain why only high doses of TMAO have an inhibitory effect on ER stress.

In figure 1 I what is the n number. Big scatter of data. Really significant?

R: n=60 in Figure 1i, and $p=0.0014$ by two-tailed Student's T test.

In figure 2 I it is difficult to see the arrow heads.

R: We increased the size of the arrowheads.

Figure 20. Pancreatic HE staining in choline diet-fed mice. Mice, male, 13 weeks of choline feeding from 8 weeks old. The arrowhead indicates inflammatory cells. Scale bar, 100 μm .

Why are figures 4 g and h displayed differently? Also, Ca^{2+} traces in h are not convincing. It is difficult to believe that there is no increase at all in Ca^{2+} subsequent to glucose stimulation in the TMAO treated group.

R: Figure 4g shows the resting cytoplasmic calcium levels under low glucose conditions. Figure 4h shows the calcium transient upon high glucose. Initially, the cells were under low glucose conditions, and cytosolic calcium was measured. Then, high glucose was added to the medium, followed by cytosolic calcium measurement again. Both Figures 4g and 4h indicate a slight increase in basal calcium in the TMAO-treated cells under low glucose conditions.

TMAO dramatically inhibited calcium transients upon high glucose stimulation in MIN6 cells. In Figure 4h of the original manuscript, the trace of vehicle group contains 43 individuals and TMAO group contains 24 individuals. So, the trace is the average of individuals. We listed some of the traces of individual cells (Figure 21). If we plotted the trace of TMAO with same scale in Y axis as the vehicle group, the TMAO group showed nearly no increase (Figures 21a and b); However, if we replotted the trace in TMAO group with a smaller scale in Y axis, the traces of TMAO group actually showed a slight increase during high glucose stimulation (Figure 21c). Moreover, when we measured ER calcium in MIN6 cells, we also applied fluorescent dye Rhod-4 to simultaneously detect cytosolic calcium in the same cells. The results showed TMAO dramatically inhibited calcium transients upon high glucose stimulation (Figure 17c).

Figure 21. The individual cells traces of vehicle (a) and TMAO (b-c) group in Figure 4h of the original manuscript. b and c showed different Y-axis.

In figure 5k how should we consider reduced Serca2 protein levels in relation to Serca2 activity?

R: Please see the responses to the question with Figure 18.

1. Krueger ES, *et al.* Gut Metabolite Trimethylamine N-Oxide Protects INS-1 β -Cell and Rat Islet Function under Diabetic Glucolipotoxic Conditions. *Biomolecules* **11**, (2021).
2. Harrison DE, Christie MR, Gray DW. Properties of isolated human islets of Langerhans: insulin secretion, glucose oxidation and protein phosphorylation. *Diabetologia* **28**, 99-103 (1985).

3. Sweet IR, *et al.* Glucose stimulation of cytochrome C reduction and oxygen consumption as assessment of human islet quality. *Transplantation* **80**, 1003-1011 (2005).
4. Doliba NM, *et al.* Glucokinase activation repairs defective bioenergetics of islets of Langerhans isolated from type 2 diabetics. *American journal of physiology Endocrinology and metabolism* **302**, E87-e102 (2012).
5. Detimary P, Dejonghe S, Ling Z, Pipeleers D, Schuit F, Henquin JC. The changes in adenine nucleotides measured in glucose-stimulated rodent islets occur in beta cells but not in alpha cells and are also observed in human islets. *J Biol Chem* **273**, 33905-33908 (1998).
6. Rorsman P, Ashcroft FM. Pancreatic β -Cell Electrical Activity and Insulin Secretion: Of Mice and Men. *Physiological reviews* **98**, 117-214 (2018).
7. Mislisler S, Gee WM, Gillis KD, Scharp DW, Falke LC. Metabolite-regulated ATP-sensitive K⁺ channel in human pancreatic islet cells. *Diabetes* **38**, 422-427 (1989).
8. Fridlyand LE, Jacobson DA, Philipson LH. Ion channels and regulation of insulin secretion in human β -cells: a computational systems analysis. *Islets* **5**, 1-15 (2013).
9. Braun M, *et al.* Voltage-gated ion channels in human pancreatic beta-cells: electrophysiological characterization and role in insulin secretion. *Diabetes* **57**, 1618-1628 (2008).
10. Mislisler S, Barnett DW, Pressel DM, Gillis KD, Scharp DW, Falke LC. Stimulus-secretion coupling in beta-cells of transplantable human islets of Langerhans. Evidence for a critical role for Ca²⁺ entry. *Diabetes* **41**, 662-670 (1992).
11. Hellman B, *et al.* Glucose induces oscillatory Ca²⁺ signalling and insulin release in human pancreatic beta cells. *Diabetologia* **37 Suppl 2**, S11-20 (1994).
12. Martín F, Soria B. Glucose-induced [Ca²⁺]_i oscillations in single human pancreatic islets. *Cell calcium* **20**, 409-414 (1996).
13. Dobbins RL, Chester MW, Stevenson BE, Daniels MB, Stein DT, McGarry JD. A fatty acid-dependent step is critically important for both glucose- and non-glucose-stimulated insulin secretion. *The Journal of clinical investigation* **101**, 2370-2376 (1998).
14. Mao GH, Chen GA, Bai HY, Song TR, Wang YX. The reversal of hyperglycaemia in diabetic mice using PLGA scaffolds seeded with islet-like cells derived from human embryonic stem cells. *Biomaterials* **30**, 1706-1714 (2009).
15. Schugar RC, *et al.* The TMAO-Producing Enzyme Flavin-Containing Monooxygenase 3 Regulates Obesity and the Beiging of White Adipose Tissue. *Cell reports* **20**, 279 (2017).

16. Bennett BJ, *et al.* Trimethylamine-N-oxide, a metabolite associated with atherosclerosis, exhibits complex genetic and dietary regulation. *Cell metabolism* **17**, 49-60 (2013).
17. Henquin JC, Ishiyama N, Nenquin M, Ravier MA, Jonas JC. Signals and pools underlying biphasic insulin secretion. *Diabetes* **51 Suppl 1**, S60-67 (2002).
18. Corkey BE, Deeney JT, Merrins MJ. What Regulates Basal Insulin Secretion and Causes Hyperinsulinemia? *Diabetes* **70**, 2174-2182 (2021).
19. Liang K, *et al.* Alterations of the Ca²⁺ signaling pathway in pancreatic beta-cells isolated from db/db mice. *Protein & cell* **5**, 783-794 (2014).
20. Miao J, *et al.* Flavin-containing monooxygenase 3 as a potential player in diabetes-associated atherosclerosis. *Nat Commun* **6**, 6498 (2015).
21. Kono T, *et al.* PPAR- γ activation restores pancreatic islet SERCA2 levels and prevents β -cell dysfunction under conditions of hyperglycemic and cytokine stress. *Molecular endocrinology (Baltimore, Md)* **26**, 257-271 (2012).
22. Esposito T, Varriale B, D'Angelo R, Amato A, Sidoti A. Regulation of flavin-containing mono-oxygenase (Fmo3) gene expression by steroids in mice and humans. *Hormone molecular biology and clinical investigation* **20**, 99-109 (2014).
23. Yabiku K, Nakamoto K, Tokushige A. Reintroducing testosterone in the db/db mouse partially restores normal glucose metabolism and insulin resistance in a leptin-independent manner. *BMC endocrine disorders* **18**, 38 (2018).
24. Zhao Y, *et al.* The Effect of Different L-Carnitine Administration Routes on the Development of Atherosclerosis in ApoE Knockout Mice. *Molecular nutrition & food research* **62**, (2018).
25. Zhang X, *et al.* Phenolamide extract of apricot bee pollen alleviates glucolipid metabolic disorders and modulates the gut microbiota and metabolites in high-fat diet-induced obese mice. *Food & function* **14**, 4662-4680 (2023).
26. Talchai C, Xuan S, Lin HV, Sussel L, Accili D. Pancreatic β cell dedifferentiation as a mechanism of diabetic β cell failure. *Cell* **150**, 1223-1234 (2012).
27. Cinti F, *et al.* Evidence of β -Cell Dedifferentiation in Human Type 2 Diabetes. *The Journal of clinical endocrinology and metabolism* **101**, 1044-1054 (2016).
28. Gao T, *et al.* Pdx1 maintains β cell identity and function by repressing an α cell program. *Cell metabolism* **19**, 259-271 (2014).

29. Khin PP, Lee JH, Jun HS. A Brief Review of the Mechanisms of β -Cell Dedifferentiation in Type 2 Diabetes. *Nutrients* **13**, (2021).
30. Tabas I, Ron D. Integrating the mechanisms of apoptosis induced by endoplasmic reticulum stress. *Nature cell biology* **13**, 184-190 (2011).
31. Reddish FN, *et al.* Rapid subcellular calcium responses and dynamics by calcium sensor G-CatchER. *iScience* **24**, 102129 (2021).
32. Byun JK, *et al.* Inhibition of Glutamine Utilization Synergizes with Immune Checkpoint Inhibitor to Promote Antitumor Immunity. *Molecular cell* **80**, 592-606.e598 (2020).
33. Bi J, *et al.* Seipin promotes adipose tissue fat storage through the ER Ca^{2+} -ATPase SERCA. *Cell metabolism* **19**, 861-871 (2014).
34. Moraru A, *et al.* THADA Regulates the Organismal Balance between Energy Storage and Heat Production. *Developmental cell* **41**, 450 (2017).
35. Achard CS, Laybutt DR. Lipid-induced endoplasmic reticulum stress in liver cells results in two distinct outcomes: adaptation with enhanced insulin signaling or insulin resistance. *Endocrinology* **153**, 2164-2177 (2012).
36. Emily S. Krueger JLB, Kacie B. Russon, Weston S. Elison, Jackson M. Hansen , Andrew P. Neilson , Jordan R. Davis, , Jason M. Hansen and Jeffery S. Tessem. Gut Metabolite Trimethylamine N-Oxide Protects INS-1 β -Cell and Rat Islet Function under Diabetic Glucolipotoxic Conditions. *Biomolecules* **11**, 1892 (2021).
37. Bui-Le L, *et al.* Revealing the complexity of ionic liquid-protein interactions through a multi-technique investigation. *Communications chemistry* **3**, 55 (2020).
38. Liao YT, Manson AC, DeLyser MR, Noid WG, Cremer PS. Trimethylamine N-oxide stabilizes proteins via a distinct mechanism compared with betaine and glycine. *Proceedings of the National Academy of Sciences of the United States of America* **114**, 2479-2484 (2017).
39. Zou Q, Bennion BJ, Daggett V, Murphy KP. The molecular mechanism of stabilization of proteins by TMAO and its ability to counteract the effects of urea. *Journal of the American Chemical Society* **124**, 1192-1202 (2002).

REVIEWER COMMENTS

Reviewer #1 (Remarks to the Author):

The authors have addressed my previous critiques.

Reviewer #2 (Remarks to the Author):

I have now had the chance to go through the authors' responses to my original critique of their manuscript.

Overall, I think they have done a good job answering the critique and amending the manuscript accordingly.

However, I am still not happy with the quality of the Ca²⁺ measurements. As far as I understand the authors have tried isolated individual normal beta cells and have had problems with the survival of these cells. I believe that it would have been more efficient to measure Ca²⁺ in cells within the intact islets.

The individual Ca²⁺ traces in figures 17 and 21 are not convincing. Maybe the authors should seek some expert help for this type of measurements.

Response to the reviewers' comments

Reviewer #1 (Remarks to the Author):

The authors have addressed my previous critiques.

Reviewer #2 (Remarks to the Author):

I have now had the chance to go through the authors' responses to my original critique of their manuscript.

Overall, I think they have done a good job answering the critique and amending the manuscript accordingly.

However, I am still not happy with the quality of the Ca²⁺ measurements. As far as I understand the authors have tried isolated individual normal beta cells and have had problems with the survival of these cells. I believe that it would have been more efficient to measure Ca²⁺ in cells within the intact islets.

The individual Ca²⁺ traces in figures 17 and 21 are not convincing. Maybe the authors should seek some expert help for this type of measurements.

R: We thank the reviewer for the encouraging words and comments. In Figure 17 of the 1st response to the reviewers' comments, the Ca²⁺ traces in normal β cells are not convincing due to the very low survival of these cells (less than 5%). As the reviewer suggested, we discussed this issue with an expert and measured Ca²⁺ in β cells within the intact primary islets from GCaMP6f mice. These mice expressed the genetically encoded calcium indicator GCaMP6f in β cells and were used for the dynamic measurement of cytosolic calcium in β cells. The results showed TMAO (100 nM, 18 h) treatment significantly reduced the peak value of calcium transients in β cells of mouse primary islets (**Figures 22a-b**).

Thanks a lot for pointing out the problems in our Fura-2 data in Figure 21 (the 1st response to the reviewers' comments). In that case, we simultaneously applied lasers at

340 and 380 nm. And these short-wavelength lasers with higher power could be a harsh stimulation for MIN6 cells. In the revised manuscript, MIN6 cells were infected with a genetically encoded GCaMP6f adenovirus (Addgene, No.67564) for calcium measurement. We could obtain better baselines and a more reliable response of cytosolic calcium triggered by high glucose (**Figure 22c**). Consistent with the finding in normal β cells, TMAO (100 nM, 18 h) treatment reduced the peak value of calcium transients in MIN6 cells (**Figure 22c**).

Figure 22. TMAO inhibited glucose-stimulated cytosolic Ca^{2+} transients in β cells of mouse primary islets and MIN6 cells. a-b Cytosolic calcium dynamics (a) and peak value of cytosolic calcium transients (b) stimulated by high glucose (25 mM) in different β cells of Ins1-GCaMP6f mouse primary islets treated with or without TMAO (100 nM, 18 h). **c** Cytosolic calcium dynamics with GCaMP6f stimulated by high glucose (16.8 mM) in MIN6 cells treated with or without TMAO (100 nM, 18 h). Error bars represent the SEM, $n=46$ in the vehicle group and $n=22$ in the TMAO group (a), $n=131$ in the vehicle group and $n=105$ in the TMAO group (b), $n=13$ in the vehicle group and $n=14$ in the TMAO group (c). Mice: Ins1-GCaMP6f mice, male, 16 weeks old (a-b). *** $p < 0.001$.